# A CURRICULUM VIEW OF ROBUST LOSS FUNCTIONS

## ABSTRACT

Robust loss functions are designed to mitigate the adverse impacts of label noise, which enjoy theoretical guarantees that is agnostic to the training dynamics. However, these guarantees fail to characterize some empirical phenomenons. We unify a broad array of loss functions into a novel standard form with a shared implicit loss function and different implicit design choices along the sample-weighting function and the output regularizer. The resulting curriculum view connects the designs of loss functions and learning curricula, leading to a straightforward analysis of the training dynamics that helps demystify existing empirical observations. In particular, we show that robust sample-weighting function sift and neglect corrupted samples. We further analyze how regularizers affect the training dynamics with different loss functions. Finally, we dissect the cause of the notorious underfitting issue and support our explanation with our effective fixes.

## 1 INTRODUCTION

Label noise is non-negligible in automatic labeling (Liu et al., 2021), crowd-sourcing (Russakovsky et al., 2015) and expert annotation (Kato & Matsubara, 2010). Their adverse impacts can be mitigated with loss functions that are theoretically robust against label noise (Song et al., 2020), which enjoys a bounded discrepancy between the optima obtained with clean or noisy labels (Ghosh et al., 2017; Zhou et al., 2021). However, existing theoretical bounds do not account for the training dynamics towards these optima. This oversight has led to confounding empirical observations, e.g., robust loss functions can underfit, or perform poorly, on difficult tasks (Zhang & Sabuncu, 2018; Wang et al., 2019c). In addition, it is unclear why seemingly contradictory regularizers (Lukasik et al., 2020; Wei et al., 2021) can improve noise-robustness at the same time.

We unify loss functions with *distinct theoretical motivations* into a standard form, unveiling their *implicit* design choices along the sample-weighting functions and the output regularizers (§3). The *shared* implicit loss function in the standard form provides a common metric to track the learning progress of each sample. We can demystify existing empirical observations by analyzing how different design choices affect the evolution of implicit loss distributions. Notably, our derivation shows that each loss function implicitly defines a learning curriculum, which specifies a sequence of training criteria at different training steps (Wang et al., 2020). This curriculum view connects the designs of loss functions and curricula that are commonly viewed as distinct approaches (Song et al., 2020).

We provide a comprehensive analysis on how different design choices affect the training dynamics, leading to plenty empirically supported *understandings* for these loss functions. We first show that sample-weighting functions of robust loss functions act as sample sifts that neglect samples with erroneous labels (§4.1), which utilizes the memorization effect (Arpit et al., 2017) similar to curriculum based approaches (Li et al., 2020). We then analyze how regularizers affect the training dynamics with different loss functions (§4.2). In particular, we are the first to show the vital role of confidence reducing regularizers, e.g., weight decay, in successful training with robust loss function. Finally, we attribute the underfitting issue of robust loss functions to minimal sample weights at initialization and support it with our effective fixes (§4.3).

## 2 RELATED WORK

Most existing studies on robust loss functions (Ghosh et al., 2017; Zhang & Sabuncu, 2018; Wang et al., 2019c; Feng et al., 2020; Liu & Guo, 2020; Cheng et al., 2021; Zhou et al., 2021) focus on deriving bounds of the difference between optima obtained with noisy and clean labels, which are agnostic to the training dynamics. We unify them into a standard form and thoroughly analyze how implicit choices along sample-weighting functions and output regularizers affect the training

dynamics. Although the underfitting issue has been heuristically mitigated (Zhang & Sabuncu, 2018; Wang et al., 2019a;c; Ma et al., 2020), we explicitly identify the cause and support it with our fixes. Both confidence reducing (Lukasik et al., 2020) and promoting (Wei et al., 2021; Cheng et al., 2021) regularizers are shown to improve the noise robustness of cross entropy. We analyze how different regularizers affect the training dynamics with different loss functions, especially with robust loss functions. We only consider loss functions with closed-form expressions and leave others (Amid et al., 2019; Wei & Liu, 2021) to future work.

Our curriculum view connects existing robust loss functions to the seemingly distinct (Song et al., 2020) curriculum learning. To achieve robustness against noisy labels, curriculum-based approaches use either sample selection (Chen et al., 2019; Huang et al., 2019) or sample weighting (Chang et al., 2017; Ren et al., 2018), typically based on the small loss trick (Song et al., 2020) utilizing the memorization effect (Arpit et al., 2017). Our work differs in three important perspectives. First, the sample weights analyzed in our work are *implicitly* defined by robust loss functions rather than *explicitly* designed (Chang et al., 2017; Wang et al., 2019a;b) or predicted by a model (Jiang et al., 2018; Ren et al., 2018). Second, we use the derived implicit loss function to track the learning of samples unlike common metrics based on loss values (Kumar et al., 2010; Loshchilov & Hutter, 2015) or gradient magnitudes (Gopal, 2016). Finally, instead of designing better sample-weighting curricula (Chang et al., 2017; Wang et al., 2019a;b), we aim to thoroughly understand how the identified design choices of *existing* loss functions affects the training dynamics. Our work is also related to the ongoing debate (Hacohen & Weinshall, 2019; Wang et al., 2020) on the strategies to select or weight samples in learning curricula: either easier first (Bengio et al., 2009; Kumar et al., 2010) or harder first (Loshchilov & Hutter, 2015; Zhang et al., 2018). The sample-weighting functions we identified in robust loss functions can be viewed as a combination of both strategies, which emphasizes samples with moderate difficulty, providing a novel weighting paradigm for curriculum learning.

Most related to our work, Wang et al. (2019a) view the L1 norm of logit gradients as sample weights and focus on an *intuitive* fix for the underfitting of MAE. We extend the analysis of logit gradients to *explicitly* identify the implicit loss function and the aforementioned design choices for *a wide range* of loss functions, leading to a unified standard form. In addition, we thoroughly analyze the training dynamics with different design choices, resulting in deeper understanding of noise robustness and vulnerability to underfitting that can facilitate better designs of loss functions and learning curricula.

## 3 DERIVING THE STANDARD FORM

We formulate classification with label noise and noise robustness before presenting our results. The $k$-ary classification task with input $\boldsymbol{x} \in \mathbb{R}^d$ can be solved by classifier $\arg\max_i s_i$, where $s_i$ is the score (logit) for the $i$-th class in function $\boldsymbol{s} : \mathbb{R}^d \to \mathbb{R}^k$ parameterized by $\boldsymbol{\theta}$. With a slight abuse of notation, we use $\boldsymbol{s}$ for both the scoring function $\boldsymbol{s}(\boldsymbol{x}; \boldsymbol{\theta})$ and its output. Given label $y^* \in \{1, \ldots, k\}$ for $\boldsymbol{x}$ in a clean data distribution $\mathcal{D}^*$ and a loss function $L(\boldsymbol{s}, y^*)$, we can estimate $\boldsymbol{\theta}$ with risk minimization $\boldsymbol{\theta}^* = \arg\min_{\boldsymbol{\theta}} \mathbb{E}_{(\boldsymbol{x},y^*) \sim \mathcal{D}^*}[L(\boldsymbol{s}(\boldsymbol{x}; \boldsymbol{\theta}), y^*)]$.

Labeling errors turn clean labels $y^*$ into noisy ones, $y \sim P(y|\boldsymbol{x}, y^*)$, leading to a noisy data distribution $\mathcal{D}$ with noise rate $\eta = P(y \neq y^*)$. A sample is *corrupted* when $y \neq y^*$. Following Ghosh et al. (2017), label noise is *symmetric* (uniform) if $P(y|\boldsymbol{x}, y^*) = \eta/(k-1), \forall y \neq y^*$ and *asymmetric* (class-conditional) when $P(y|\boldsymbol{x}, y^*) = P(y|y^*)$. Loss function $L$ is robust against label noise if parameters estimated with $\mathcal{D}$, $\boldsymbol{\theta}' = \arg\min_{\boldsymbol{\theta}} \mathbb{E}_{(\boldsymbol{x},y) \sim \mathcal{D}}[L(\boldsymbol{s}(\boldsymbol{x}; \boldsymbol{\theta}), y)]$, leads to bounded extra risk

$$\mathbb{E}_{(\boldsymbol{x},y^*) \sim \mathcal{D}^*}[L(\boldsymbol{s}(\boldsymbol{x}; \boldsymbol{\theta}'), y^*)] - \mathbb{E}_{(\boldsymbol{x},y^*) \sim \mathcal{D}^*}[L(\boldsymbol{s}(\boldsymbol{x}; \boldsymbol{\theta}^*), y^*)] \leq \epsilon, \tag{1}$$

under clean data distribution $(\boldsymbol{x}, y^*) \sim \mathcal{D}^*$, where $\epsilon$ is a constant.

We examine a broad array of loss functions with closed-form expressions in this work, among which Cross Entropy (CE), Focal Loss (FL; Lin et al., 2017) and Symmetric Cross Entropy (SCE; Wang et al., 2019c) are not robust as they have unbounded extra risk. In contrast, Mean Absolute Error (MAE; Ghosh et al., 2017), Taylor Cross Entropy (TCE; Feng et al., 2020), Generalized Cross Entropy (GCE; Zhang & Sabuncu, 2018), Mean Square Error (MSE; Ghosh et al., 2017), Peer Loss (PL; Liu & Guo, 2020) and asymmetric losses (Zhou et al., 2021) including Asymmetric Generalized Cross Entropy (AGCE), Asymmetric Unhinged Loss (AUL) and Asymmetric Exponential Loss (AEL) are all robust. See Appendix A.2 for a more thorough review and the derivation of equivalence between the Reverse Cross Entropy (RCE; Wang et al., 2019c) and MAE.

| Name | Formula with $\tilde{L}(p_y)$ | $w(\boldsymbol{s}, y)$ | Constraints |
|------|------|------|------|
| CE | $-\log p_y$ | $1 - p_y$ | |
| FL | $-(1 - p_y)^q \log p_y$ | $(1 - p_y)^q (1 - p_y - q p_y \log p_y)$ | $q > 0$ |
| SCE | $-(1 - q) \log p_y + q(1 - p_y)$ | $(1 - q + q \cdot p_y)(1 - p_y)$ | $0 < q < 1$ |
| TCE | $\sum_{i=1}^{q} (1 - p_y)^i / i$ | $p_y \sum_{i=1}^{q} (1 - p_y)^i$ | $q \in \mathbb{N}^+$ |
| GCE | $(1 - p_y^q)/q$ | $p_y^q (1 - p_y)$ | $0 < q \le 1$ |
| RCE / MAE | $1 - p_y$ | $p_y(1 - p_y)$ | |
| AEL | $e^{-p_y/q}$ | $p_y(1 - p_y)e^{-p_y/q}/q$ | $q > 0$ |
| AUL | $[(a - p_y)^q - (a - 1)^q]/q$ | $p_y(1 - p_y)(a - p_y)^{q-1}$ | $a > 1, q > 0$ |
| AGCE | $[(a + 1)^q - (a + p_y)^q]/q$ | $p_y(1 - p_y)(a + p_y)^{q-1}$ | $a > 0, q > 0$ |

Table 1: Formulae, hyperparameter constraints and sample-weighting functions $w(\boldsymbol{s}, y)$ for loss functions conforming to Eq. (3). See Appendix A.2 for plots of $w(\boldsymbol{s}, y)$. Notably, $w(\boldsymbol{s}, y)$ of all robust loss functions emphasize samples with moderate $\Delta(\boldsymbol{s}, y)$.

We now derive the standard form of the aforementioned loss functions. As shown in Table 1, most loss functions are functions of the softmax probability, $L(\boldsymbol{s}, y) = \tilde{L}(p_y)$, where

$$p_y = \frac{e^{s_y}}{\sum_i e^{s_i}} = \frac{1}{e^{-(s_y - \log \sum_{i \neq y} e^{s_i})} + 1} = \frac{1}{e^{-\Delta(\boldsymbol{s}, y)} + 1},$$

with

$$\Delta(\boldsymbol{s}, y) = s_y - \log \sum_{i \neq y} e^{s_i} \le s_y - \max_{i \neq y} s_i \tag{2}$$

measuring the score margin between the labeled class $y$ and any other classes. Since $\nabla_{\boldsymbol{s}}\Delta(\boldsymbol{s}, y)$ has a constant norm, $\|\nabla_{\boldsymbol{s}}\Delta(\boldsymbol{s}, y)\|_1 = 2$, extracting it from the sample gradient $\nabla_{\boldsymbol{s}}L(\boldsymbol{s}, y)$

$$\nabla_{\boldsymbol{s}}L(\boldsymbol{s}, y) = \frac{\mathrm{d}\tilde{L}(p_y)}{\mathrm{d}\Delta(\boldsymbol{s}, y)} \cdot \nabla_{\boldsymbol{s}}\Delta(\boldsymbol{s}, y) = -w(\boldsymbol{s}, y) \cdot \nabla_{\boldsymbol{s}}\Delta(\boldsymbol{s}, y)$$

factorizes the scale $w(\boldsymbol{s}, y)$ and direction $\nabla_{\boldsymbol{s}}\Delta(\boldsymbol{s}, y)$. After detaching $w(\boldsymbol{s}, y)$ from derivative and integral and integrating over $\boldsymbol{s}$, *equivalent under first-order optimizers*, $L(\boldsymbol{s}, y)$ can be rewritten into

$$L'(\boldsymbol{s}, y) = -w(\boldsymbol{s}, y) \cdot \Delta(\boldsymbol{s}, y) \tag{3}$$

where $w(\boldsymbol{s}, y) = -\kappa[\mathrm{d}\tilde{L}(p_y)/\mathrm{d}\Delta(\boldsymbol{s}, y)] > 0$ is a **sample-weighting function** with $\kappa(\cdot)$ the detach operator and $\Delta(\boldsymbol{s}, y)$ serves as a shared **implicit loss function**. We show $w(\boldsymbol{s}, y)$ of different loss functions in Table 1 and their plots in Fig. 5 of Appendix A. Intuitively, $\Delta(\boldsymbol{s}, y)$ determines the gradient directions to learn the labeled samples, while $w(\boldsymbol{s}, y)$ affects the priority of sample learning by emphasizing different samples during training. Loss functions conforming to Eq. (3) thus implicitly define different sample-weighting curricula. Unlike loss values (Loshchilov & Hutter, 2015) and gradient magnitudes (Gopal, 2016), $\Delta(\boldsymbol{s}, y)$ is *shared* by loss functions considered in this work and isolates the influence of $w(\boldsymbol{s}, y)$ and arbitrary loss scales. It further avoids the sigmoid transform in $p_y$ rendering the change of large $|\Delta(\boldsymbol{s}, y)|$ less perceptible. $\Delta(\boldsymbol{s}, y)$ is thus an effective common metric for tracking the learning progress of samples with different loss functions.

Loss functions additionally depending on $\{p_i\}_{i \neq y}$ may fail to conform to Eq. (3), including Normalized Cross Entropy (NCE; Ma et al., 2020), Mean Square Error (MSE; Ghosh et al., 2017), Peer Loss (PL; Liu & Guo, 2020), Jensen-Shannon Divergence (JS; Englesson & Azizpour, 2021), and CE with Confidence Regularization (CR; Cheng et al., 2021), Negative Label Smoothing (NLS; Wei et al., 2021) or Label Smoothing (LS; Lukasik et al., 2020). However, we can extract an output regularizer $R(\boldsymbol{s})$ constraining the distribution of $\boldsymbol{s}$, leaving a **primary loss function** $L'(\boldsymbol{s}, y)$ conforming to Eq. (3). We show the derivation of NCE and leave other loss functions to Appendix A.5. Given

$$\nabla_{\boldsymbol{s}}L_{\mathrm{NCE}}(\boldsymbol{s}, y) = \nabla_{\boldsymbol{s}} \frac{L_{\mathrm{CE}}(\boldsymbol{s}, y)}{\sum_i L_{\mathrm{CE}}(\boldsymbol{s}, i)} = \frac{-1}{\sum_{i=1}^{k} \log p_i} \nabla_{\boldsymbol{s}} L_{\mathrm{CE}}(\boldsymbol{s}, y) + \frac{-k \log p_y}{(\sum_{i=1}^{k} \log p_i)^2} \nabla_{\boldsymbol{s}} \sum_{i=1}^{k} \frac{1}{k} \log p_i$$

| Name | Formula with $p_i$ | Primary | $R(\boldsymbol{s})$ | $\Delta(\boldsymbol{s}^*, y)$ |
|------|-------------------|---------|---------------------|-------------------------------|
| NCE | $\log p_y / \sum_{i=1}^{k} \log p_i$ | $-\gamma \cdot \log p_i$ | $\sum_{i=1}^{k} \frac{1}{k} \log p_i$ | $\pm\infty$ |
| MSE | $\sum_{i=1}^{k} [\mathbb{I}(i = y) - p_i]^2$ | $1 - p_y$ | $\sum_{i=1}^{k} p_i^2$ | $-\log k$ |
| PL (CE + CR) | $-\log p_y + \mathbb{E}_{y \sim \mathcal{D}, \mathbf{x} \sim \mathcal{D}}[\log p_{y\|\mathbf{x}}]$ | $-\log p_y$ | $\sum_{i=1}^{k} P(y = i) \log p_i$ | $\pm\infty$ |
| JS | $a \sum_i p_i \log \frac{p_i}{m_i} + (1 - a) \sum_i e_i \log \frac{e_i}{m_i}$ | $\tilde{L}'_{\mathrm{JS}}(p_y)$ | $\sum_{i=1}^{k} p_i$ | $-\log k$ |
| CE + LS | $-\sum_{i=1}^{k} [\mathbb{I}(i = y)(1 - q) + \frac{q}{k}] \log p_i$ | $-\log p_y$ | $-\sum_{i=1}^{k} \frac{1}{k} \log p_i$ | $-\log k$ |
| CE + NLS | $-\sum_{i=1}^{k} [\mathbb{I}(i = y)(1 + r) - \frac{r}{k}] \log p_i$ | $-\log p_y$ | $\sum_{i=1}^{k} \frac{1}{k} \log p_i$ | $\pm\infty$ |

Table 2: Formulae, primary loss functions, output regularizers and $\Delta(\boldsymbol{s}^*, y)$ at the minimum $\boldsymbol{s}^*$ of the regularizers for loss functions conforming to Eq. (4), where $0 < a < 1$, $0 \leq q < 1$ and $r > 0$ are all hyperparameters. We view PL in its expectation to derive its regularizer CR (Cheng et al., 2021). $p_{y\|\mathbf{x}}$ is the softmax probability of a random label y with a random input $\mathbf{x}$ sampled from the noisy data distribution. For JS, $e_i = \mathbb{I}(i = y)$ where $\mathbb{I}(\cdot)$ is the indicator function and $m_i = ap_i + (1-a)e_i$. Finally, $\tilde{L}'_{\mathrm{JS}}(p_y) = ap_y \log ap_y - (ap_y + 1 - a)\log(ap_y + 1 - a)$.

by detaching weighting functions from derivative and integral and integrating over $\boldsymbol{s}$, NCE becomes

$$L''_{\mathrm{NCE}}(\boldsymbol{s}, y) = \gamma(\boldsymbol{s}, y) \cdot w_{\mathrm{CE}}(\boldsymbol{s}, y) \cdot \Delta(\boldsymbol{s}, y) + \lambda_{\mathrm{NCE}}(\boldsymbol{s}, y) \cdot R_{\mathrm{NCE}}(\boldsymbol{s})$$

where $\gamma(\boldsymbol{s}, y) = \kappa(-1/\sum_{i=1}^{k} \log p_i)$ and $\lambda_{\mathrm{NCE}}(\boldsymbol{s}, y) = \kappa[-k \log p_y/(\sum_{i=1}^{k} \log p_i)^2]$ are weighting functions wrapped with the detach operator $\kappa(\cdot)$, and $R_{\mathrm{NCE}}(\boldsymbol{s}) = \sum_{i=1}^{k} \frac{1}{k} \log p_i$ is the output regularizer of NCE. Similar derivations thus lead to a **standard form** of loss functions,

$$L''(\boldsymbol{s}, y) = L'(\boldsymbol{s}, y) + \lambda(\boldsymbol{s}, y) \cdot R(\boldsymbol{s}) = -w(\boldsymbol{s}, y) \cdot \Delta(\boldsymbol{s}, y) + \lambda(\boldsymbol{s}, y) \cdot R(\boldsymbol{s}), \tag{4}$$

where $\lambda(\boldsymbol{s}, y)$ is a weighting function adjusting the per-sample strength of regularization, which is typically a constant when explicit regularization are applied, e.g., label smoothing. Intuitively, the output regularizer $R(\boldsymbol{s})$ constrains the distributions of $\Delta(\boldsymbol{s}, y)$ towards a predefined optimum $\Delta(\boldsymbol{s}^*, y)$, where $\boldsymbol{s}^* = \arg\min_{\boldsymbol{s}} R(\boldsymbol{s})$. We show loss functions conforming to Eq. (4), their primary loss functions and regularizers, and $\Delta(\boldsymbol{s}^*, y)$ of regularizers in Table 2.

Our derivation based on $\nabla_{\boldsymbol{s}} L(\boldsymbol{s}, y)$ is inspired by Wang et al. (2019a), who view $\|\nabla_{\boldsymbol{s}} L(\boldsymbol{s}, y)\|_1$ as sample weights for MAE and CE. We further use our derivative-detach-integral trick to *explicitly* extract the orthogonal components $w(\boldsymbol{s}, y)$, $\Delta(\boldsymbol{s}, y)$ and $R(\boldsymbol{s})$ from a wide range of loss functions, which unifies them into a standard form and identifies their implicit differences in $w(\boldsymbol{s}, y)$ and $R(\boldsymbol{s})$.

## 4 TRAINING DYNAMICS OF ROBUST LOSS FUNCTIONS

Based on the curriculum view with the standard form Eq. (4), we analyze the training dynamics of loss functions from random initializations towards the optima. Ideal loss functions should facilitate rapid learning while suppress the learning of corrupted samples. We analyze how implicit design choices along the sample-weighting function $w(\boldsymbol{s}, y)$ and the output regularizers $R(\boldsymbol{s})$ affect the evolution of $\Delta(\boldsymbol{s}, y)$ distributions, which provide a fine-grained view of the learning progress. We mainly use MAE and CE for illustration as they exhibit typical empirical observations. Analysis of other loss functions with similar results are left to Appendix B.

We conduct experiments on CIFAR10/100 (Krizhevsky, 2009) with synthetic symmetric and asymmetric label noise following Ma et al. (2020); Zhou et al. (2021), as well as human label noise (Wei et al., 2022). For symmetric label noise, the labels are randomly flipped to a different class. For asymmetric label noise on CIFAR10, we randomly flip TRUCK → AUTOMOBILE, BIRD → AIRPLANE, DEER → HORSE, CAR ↔ DOG. For CIFAR100, the 100 classes are grouped into 20 super-classes and labels are flipped within the same super-class into the next in a circular fashion. Human label noise is adopted from (Wei et al., 2022). To reflect more difficult settings than those in existing research, we include results on the large scale noisy dataset WebVision (Li et al., 2017) with *larger* subsets of classes. We use momentum 0.9, learning rate 0.1, batch size 128 and weight decay 5e-4 across all settings. We adopt an 8-layer CNN for CIFAR10, a ResNet-34 for CIFAR100 and a ResNet50 for WebVision, respectively, all with batch normalization. We train for 120, 200

| | Clean | Asymmetric | | | Symmetric | | | | | | | Human | |
| | | $\eta = 0.2$ | | $\eta = 0.2$ | | $\eta = 0.4$ | | $\eta = 0.8$ | | | | $\eta = 0.4$ | |
| Loss | Acc | $\Delta_{\mathrm{acc}}$ | $\rho$ | $\Delta_{\mathrm{acc}}$ | $\rho$ | $\Delta_{\mathrm{acc}}$ | $\rho$ | $\Delta_{\mathrm{acc}}$ | $\rho$ | | | $\Delta_{\mathrm{acc}}$ | $\rho$ |
|---|---|---|---|---|---|---|---|---|---|---|---|---|---|
| CE | 93.28 | -6.79 | 74.24 | -14.67 | 61.30 | -29.54 | 47.38 | -65.21 | 19.75 | | | -27.73 | 44.70 |
| FL | 92.88 | -3.31 | 77.33 | -12.55 | 57.29 | -27.20 | 43.42 | -65.50 | 19.52 | | | -25.64 | 41.08 |
| SCE | 93.09 | -5.84 | 79.62 | -6.72 | 74.02 | -16.03 | 60.69 | -55.27 | 23.39 | | | -21.26 | 53.58 |
| AEL | 92.59 | -2.03 | 90.00 | -2.00 | 93.89 | -4.69 | 84.88 | -39.04 | 34.72 | | | -11.15 | 66.41 |
| AUL | 92.52 | -2.41 | 89.97 | -1.83 | 93.66 | -4.59 | 84.41 | -35.25 | 36.30 | | | -12.49 | 65.95 |
| GCE | 92.51 | -1.99 | 90.92 | -2.12 | 93.91 | -4.98 | 84.92 | -31.80 | 33.45 | | | -11.21 | 67.99 |
| TCE | 92.67 | -2.51 | 89.84 | -2.12 | 93.65 | -4.62 | 84.51 | -35.08 | 36.79 | | | -12.13 | 66.32 |
| MAE | 92.25 | -10.60 | 87.64 | -1.70 | 93.95 | -4.43 | 84.41 | -82.25 | 21.60 | | | -22.18 | 65.89 |
| AGCE | 92.23 | -20.27 | 85.95 | -2.36 | 93.35 | -3.99 | 83.20 | -82.23 | 20.66 | | | -29.75 | 64.37 |

Table 3: Robust loss functions assign larger weights to clean samples. We report drop in accuracy $\Delta_{\mathrm{acc}}$ and proportion of cumulative weights for clean samples $\rho$ on CIFAR10 averaged with 3 different runs. Hyperparameters tuned with symmetric noise $\eta = 0.4$ are listed in Table 7 of Appendix B.

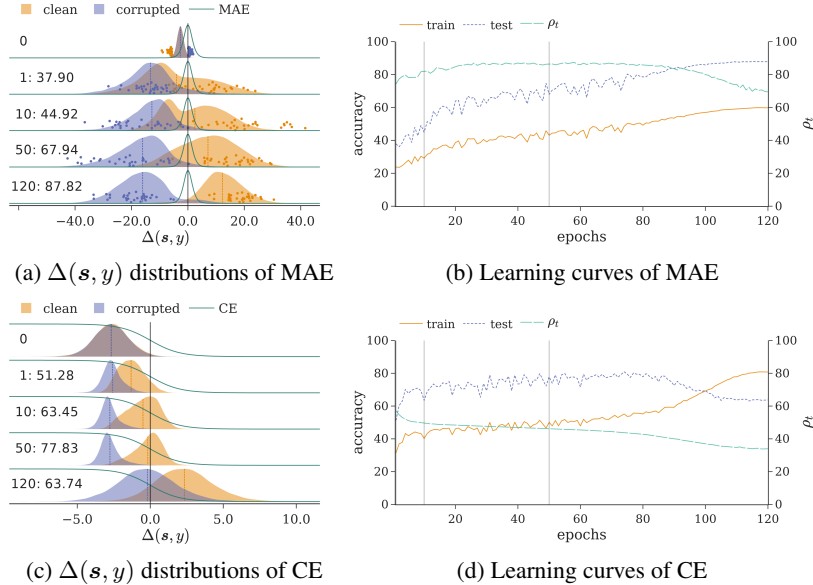

(a) $\Delta(\boldsymbol{s}, y)$ distributions of MAE

(b) Learning curves of MAE

(c) $\Delta(\boldsymbol{s}, y)$ distributions of CE

(d) Learning curves of CE

Figure 1: Training dynamics of MAE and CE on CIFAR10 with symmetric noise $\eta = 0.4$. Points of the top 1% and bottom 1% $\Delta(\boldsymbol{s}, y)$ of corrupted and clean samples, respectively, are included to demonstrate the effect of generalization in (a). We mark the median line of each distribution and include test accuracies in (a, c). Epochs sampled in (a, c) are marked in (b, d).

and 250 epochs on CIFAR10, CIFAR100 and WebVision, respectively. Unlike standard settings, we scale all $w(\boldsymbol{s}, y)$ to *unit maximum* to avoid complications, since hyperparameters of loss functions can change the scale of $w(\boldsymbol{s}, y)$, essentially adjusting the learning rate of SGD.

## 4.1 EFFECTS OF SAMPLE-WEIGHTING FUNCTIONS

We show that $w(\boldsymbol{s}, y)$ of robust loss functions implicitly implement a sample sift that neglects corrupted samples by utilizing the memorization effect (Arpit et al., 2017), i.e., models tend to learn simple and general patterns before overfitting to noisy patterns. In addition, we show that the shared monotonically decreasing right tail of $w(\boldsymbol{s}, y)$ for bounded parameter update is responsible to the eventual shift into a noise overfitting phase observed in the memorization effect.

We first show that robust loss functions assign larger weights to clean samples, which leads to better noise robustness. The emphasis on clean samples can be estimated with the ratio of their cumulative weights, $\rho_t = \sum_i \mathbb{I}(y_i = y_i^*) w_{i,t} / \sum_i w_{i,t}$, where $w_{i,t}$ is the weight of the $i$-th sample at epoch $t$ and $\mathbb{I}(\cdot)$ is the indicator function. The weighted-averaged $\rho_t$ with learning rate $\alpha_t$, $\rho = \sum_t \alpha_t \rho_t / \sum_t \alpha_t$,

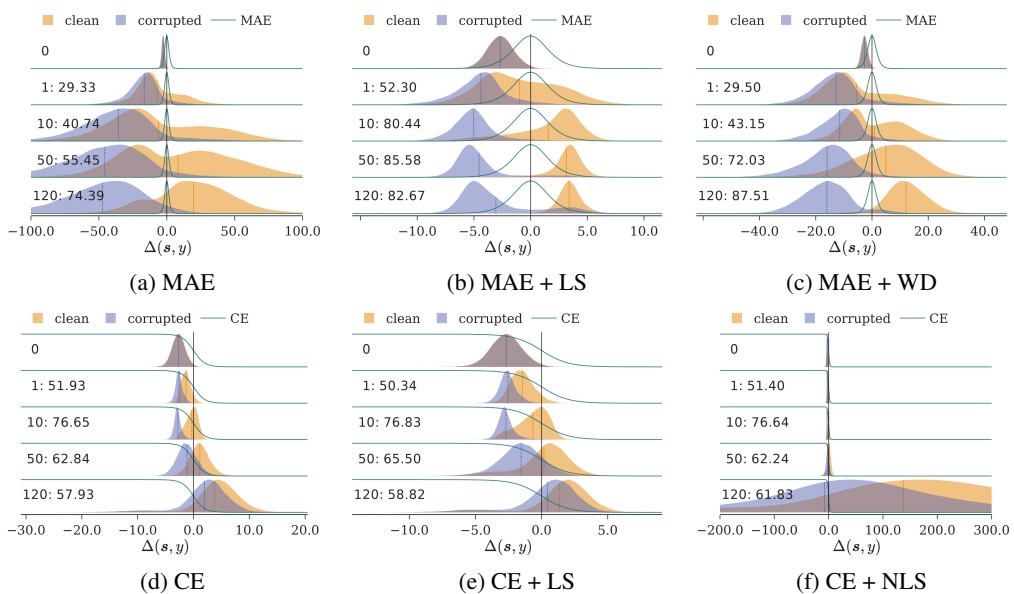

Figure 2: Training dynamics with various loss functions and regularizers on CIFAR10 with symmetric noise $\eta = 0.4$. We mark the median (dashed lines) of each distribution, and include the test accuracy of each sampled epoch.

reflects the overall emphasis of clean samples during training. In Table 3, loss functions with robustness guarantees have higher $\rho$ in general. In addition, higher $\rho$ generally corresponds to less performance drop under label noise, i.e., being more robust empirically.

A monotonically increasing left tail of $w(s, y)$ sifts samples via the memorization effect. Learning is determined by gradients estimated using samples with nontrivial weights. In Fig. 1a, $\Delta(s, y)$ of samples are randomly initialized around the left tail of $w_{\text{MAE}}(s, y)$. The similar or random gradient directions among clean or corrupted samples enhance or cancel each other, respectively Chatterjee & Zielinski (2022), leading to the assimilation of generalizable patterns during the early learning stage, when clean samples quickly improves and receive more weights, which facilitate even faster learning. Conversely, corrupted samples are unlearned due to generalization from clean samples and improve slowly with lower sample weights. As $\lim_{\Delta(s,y) \to -\infty} w_{\text{MAE}}(s, y) = 0$, unlearned samples can be ignored with minimal weights. Unlike $w_{\text{MAE}}(s, y)$, in Fig. 1c, $w_{\text{CE}}(s, y)$ emphasizes samples that are not well-learned, typically the corrupted ones, thus risk overfitting as shown in Fig. 1d. See Fig. 6 and 7 in Appendix B for similar results with different loss functions and noise settings.

Sample-weighting functions of the loss functions considered share a monotonically decreasing right tail with $\lim_{\Delta(s,y) \to \infty} w(s, y) = 0$. Although it prevents unbounded updates of well-learned samples and ensures the convergence of gradient descent training, it is also responsible for the shift into the noise overfitting stage. Despite initial dominance of the expected gradient, the faster learning clean samples will eventually receive minimal weights. Corrupted samples then come into dominance due to their relatively larger weights, when further training leads to overfitting, as shown by the synchronous drop of $\rho_t$ and test accuracies in Fig. 1d. Such explanation complements the theoretical analysis for binary linear classification in Liu et al. (2020), which reach similar conclusion that corrupted samples dominate the expected gradients in the late training stage. Regularizers preventing the overfitting phase, such as early stopping (Song et al., 2019), label smoothing (Lukasik et al., 2020) or temporal ensembling (Liu et al., 2020), can thus improve noise robustness.

## 4.2 EFFECTS OF REGULARIZATION

We examine how the extracted regularizers in Table 2 help noise-robust learning with different loss functions. They either promote (with $\Delta(s^*, y) = \pm\infty$) or reduce (with $\Delta(s^*, y) = 0$) the prediction confidence $p_y$. We show benefits of proper regularization when using robust loss functions, and further resolve the apparent contradiction that they both improve noise robustness of CE (Lukasik et al.,

|  | CIFAR10 |  |  |  |  |  | CIFAR100 |  |  |  |  |  |
|---|---|---|---|---|---|---|---|---|---|---|---|---|
|  | Clean $\eta = 0$ |  |  | Symmetric $\eta = 0.4$ |  |  | Clean $\eta = 0$ |  |  | Symmetric $\eta = 0.4$ |  |  |
| Loss | Train | Test | $\alpha^*$ | Train | Test | $\alpha^*$ | Train | Test | $\alpha^*$ | Train | Test | $\alpha^*$ |
| CE | 99.96 | 93.24 | 2.27 | 80.05 | 65.66 | 5.48 | 99.96 | 77.36 | 5.37 | 99.89 | 48.45 | 7.19 |
| SCE | 99.96 | 93.74 | 2.35 | 80.29 | 64.01 | 5.61 | 99.96 | 77.21 | 5.53 | 99.88 | 48.50 | 7.24 |
| FL | 99.89 | 93.08 | 1.94 | 80.14 | 66.14 | 5.38 | 99.93 | 77.70 | 5.08 | 99.84 | 52.87 | 6.71 |
| AEL | 99.88 | 92.97 | 2.01 | 77.05 | 67.31 | 3.75 | 99.51 | 76.75 | 2.05 | 67.92 | 62.07 | 1.63 |
| AGCE | 99.70 | 93.33 | 1.88 | 66.21 | 83.80 | 2.44 | 94.71 | 72.37 | 1.76 | 61.88 | 62.43 | 1.40 |
| AUL | 99.90 | 93.49 | 1.99 | 75.51 | 69.39 | 3.39 | 99.11 | 75.67 | 1.74 | 65.30 | 61.18 | 1.41 |
| GCE | 99.33 | 92.82 | 1.43 | 63.16 | 86.62 | 1.60 | 97.39 | 74.51 | 1.55 | 61.36 | 64.07 | 1.20 |
| TCE | 99.61 | 92.82 | 1.63 | 64.76 | 84.80 | 1.94 | 71.42 | 57.95 | 0.87 | 39.50 | 43.09 | 0.74 |
| MAE | 98.00 | 92.00 | 0.68 | 59.81 | 87.61 | 0.52 | 8.56 | 8.25 | 0.06 | 3.47 | 4.70 | 0.08 |

Table 4: Robust loss functions can underfit CIFAR100 but CIFAR10. We report training and test accuracies and $\alpha^* = \sum_t \alpha_t^*$ (scaled by 1000) at the final step under different noise settings averaged with 3 different runs. Hyperparameters tuned on CIFAR100 with symmetric noise $\eta = 0.4$ are listed in Table 7 in Appendix B.

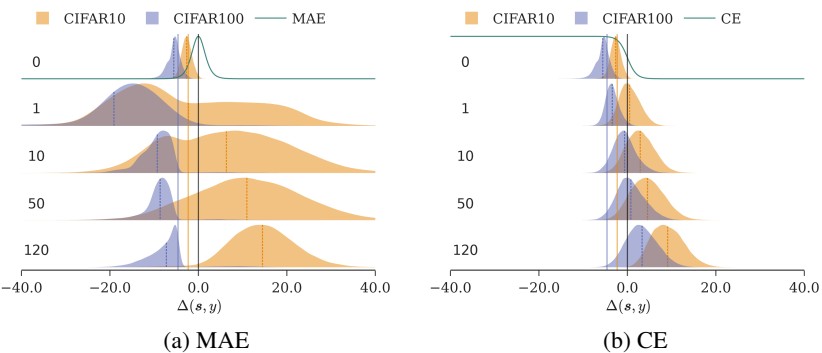

|  |  |
|---|---|
| (a) MAE | (b) CE |

Figure 3: Training dynamics of clean CIFAR100 and CIFAR10 with MAE and CE. We mark the median (dashed line) of $\Delta(\boldsymbol{s}, y)$ distributions and $\Delta(\boldsymbol{s}, y) = -\log k$ (solid line).

2020; Wei et al., 2021; Cheng et al., 2021). As weight decay constrains $\Delta(\boldsymbol{s}, y)$ towards $-\log k$, it is disabled in this section except explicitly noted. We present results with typical regularizers on CIFAR10 and leave additional results to Fig. 8 in Appendix B.

Confidence reducing regularizers can improve convergence rate and training stability[1] of robust loss functions. In Fig. 2a, a random subset of samples are assign with large weights at initialization with $w_{\mathrm{MAE}}(\boldsymbol{s}, y)$. With a high noise rate, the false generalization from these faster learning samples keep moving other samples into the low weight region, resulting in much spread and mixed $\Delta(\boldsymbol{s}, y)$ distributions of clean and corrupted samples. In contrast, with confidence reducing regularizers, samples get dragged towards $\Delta(\boldsymbol{s}, y) = -\log 10$ that corresponds to moderate sample weights, helping the accidentally unlearned samples move to high weight region and preventing overfitting faster learning samples, leading to compact and well separated $\Delta(\boldsymbol{s}, y)$ distributions in Fig. 2b and 2c.

Both types of regularizers abate the emphasis of $w_{\mathrm{CE}}(\boldsymbol{s}, y)$ on slow-learning samples, typically the corrupted ones with peculiar patterns. In Fig. 2e, confidence-reducing regularizers shrink $\Delta(\boldsymbol{s}, y)$ towards $-\log k$, hampering the learning of corrupted samples. Conversely, in Fig. 2f, confidence-promoting regularizers preserve the generalization from the faster learning clean samples, preventing the learning of corrupted samples from distorting the correct generalization. Such explanation complements the intuition that increasing prediction confidence improves noise robustness (Wei et al., 2021; Cheng et al., 2021). However, as confidence promoting regularizers lead to unbounded parameter updates, they should be combined with early stopping to avoid numerical overflow.

---

[1]We observe that MAE without regularization can underfit on CIFAR10 with $\eta = 0.8$ under some random initializations. In contrast, all experiments with proper weight decay stably achieve similars performance.

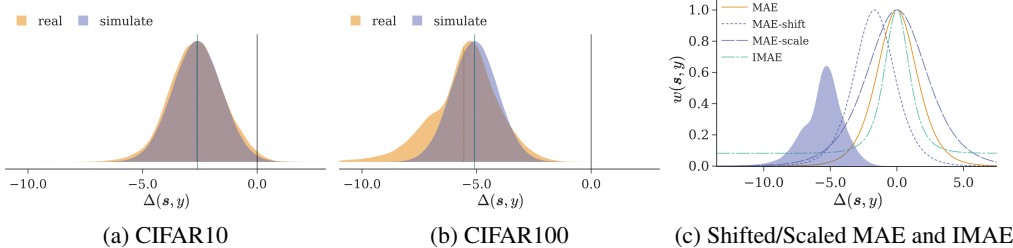

(a) CIFAR10      (b) CIFAR100      (c) Shifted/Scaled MAE and IMAE

Figure 4: (a-b): comparisons between the simulated $\Delta(\boldsymbol{s}, y)$ with $s_i \sim \mathcal{N}(0, 1)$ and real $\Delta(\boldsymbol{s}, y)$ distributions at random initialization. The estimated $\mathbb{E}_k[\Delta(\boldsymbol{s}, y)]$ is marked with green vertical lines. (c): $w_{\text{IMAE}}$ and the shifted and scaled $w_{\text{MAE}}$ of MAE with hyperparamters $T = 10$ and $\tau = 3.4$. The $\Delta(\boldsymbol{s}, y)$ distribution of CIFAR100 at random initialization is included for reference.

## 4.3 DISSECTING THE UNDERFITTING ISSUE

Robust loss functions are notorious for underfitting difficult tasks (Song et al., 2020). By comparing MAE with CE, Zhang & Sabuncu (2018) attribute underfitting of MAE to the lack of the $1/p_y$ term in sample gradients, which "treats every sample equally" and thus hampers learning. In contrast, we show that MAE emphasizes samples with moderate $\Delta(\boldsymbol{s}, y)$ which is also observed by Wang et al. (2019a). Ma et al. (2020) attribute underfitting to failure in balancing the active ($\forall i \neq y, \ l(\boldsymbol{s}, i) = 0$) and passive ($\exists i \neq y, \ l(\boldsymbol{s}, i) \neq 0$) parts of loss functions $L(\boldsymbol{s}, y) = \sum_{i=1}^{k} l(\boldsymbol{s}, i)$. However, ambiguities arise when specifying $l(\boldsymbol{s}, i)$: given $L_{\text{MAE}}(\boldsymbol{s}, y) \propto \sum_i |\mathbb{I}(i = y) - p_i| \propto \sum_i \mathbb{I}(i = y)(1 - p_i)$ where $\mathbb{I}(\cdot)$ is the indicator function, MAE can be active with $l(\boldsymbol{s}, i) = \mathbb{I}(i = y)(1 - p_i)$ but passive with $l(\boldsymbol{s}, i) = |\mathbb{I}(i = y) - p_i|$. Wang et al. (2019a) view $\|\nabla_{\boldsymbol{s}} L(\boldsymbol{s}, y)\|_1$ as weights for sample gradients and argue that their low variance makes informative and uninformative samples less distinguishable. However, it is unclear how low variance of gradient magnitudes leads to underfitting.

Alternatively, for loss functions conforming to Eq. (3), we show that underfitting results from minimal sample weights at initialization. The scale of parameter updates

$$\|\boldsymbol{\theta}_t - \boldsymbol{\theta}_{t-1}\|_p \leq \|\boldsymbol{\theta}_t - \boldsymbol{\theta}_{t-1}\|_1 \leq \alpha_t \cdot \|\nabla_{\boldsymbol{s}} L(\boldsymbol{s}, y)\|_1 \cdot \|\nabla_{\boldsymbol{\theta}} \boldsymbol{s}(\boldsymbol{x}; \boldsymbol{\theta})\|_1 = 2\alpha_t w(\boldsymbol{s}, y) \cdot \|\nabla_{\boldsymbol{\theta}} \boldsymbol{s}(\boldsymbol{x}; \boldsymbol{\theta})\|_1,$$

is affected by the effective learning rate $\alpha_t^* = 2\alpha_t w(\boldsymbol{s}, y)$ with various loss functions. Loss functions that underfit correspond to negligible cumulative effective learning rate $\alpha^* = \sum_t \alpha_t^*$ in Table 4. As shown in Fig. 3, since CIFAR100 has much smaller $\Delta(\boldsymbol{s}, y)$ at initialization, with robust loss functions like MAE, only a limited random proportion of samples receive noneligible weights, leading to a less representative estimation of the expected gradient. The majority of samples thus get stuck in the region of low sample weights. Weight decay helps little as $\Delta(\boldsymbol{s}^*, y) = -\log 100$ corresponds to negligible sample weights. These effects add up to the underfitting of MAE on CIFAR100.

The underfitting issue on CIFAR100 instead of CIFAR10 has been vaguely attributed to the increased difficulty of CIFAR100. We further dissect what makes a task difficult. In particular, the increased number of classes leads to smaller expected $\Delta(\boldsymbol{s}, y)$ at initialization. Assume that class scores $s_i$ are i.i.d. Gaussian variables $s_i \sim \mathcal{N}(\mu, \sigma)$ at initialization. Typically, we have $\mu = 0$ and $\sigma = 1$ with standard settings (Glorot & Bengio, 2010; He et al., 2015; Ioffe & Szegedy, 2015). The expected $\Delta(\boldsymbol{s}, y)$ with number of classes $k$ can be approximated with

$$\mathbb{E}_k[\Delta(\boldsymbol{s}, y)] \approx -\log(k-1) - \sigma^2/2 + \frac{e^{\sigma^2} - 1}{2(k-1)} \tag{5}$$

where $\mathbb{E}_k[\Delta(\boldsymbol{s}, y)] < 0$ with $\sigma \approx 1$ and a large $k$, e.g., $k = 100$. See Fig. 4a and 4b for a comparison between the simulated and real $\Delta(\boldsymbol{s}, y)$ distributions at random initialization. We leave derivations and more comparisons between our assumptions and real settings to Appendix B. Combined with the decreased $\Delta(\boldsymbol{s}^*, y) = -\log k$ of the confidence reducing regularizers, initial learning with $\lim_{\Delta(\boldsymbol{s}, y) \to -\infty} w(\boldsymbol{s}, y) = 0$ can get stuck with minimal sample weights, thus leading to underfitting.

Our analysis suggests that the fixed sample-weighting function $w(\boldsymbol{s}, y)$ is to blame for underfitting. We can thus morph the weighting functions for nontrivial initial sample weights, which can be simply achieved by scaling

$$w^*(\boldsymbol{s}, y) = w^*(\Delta(\boldsymbol{s}, y)) = w(\Delta(\boldsymbol{s}, y)/|\mathbb{E}_k[\Delta(\boldsymbol{s}, y)]| \cdot \tau)$$

| Loss | Clean $\eta = 0$ | Symmetric $\eta = 0.4$ | $\eta = 0.8$ | Asymmetric $\eta = 0.4$ | Human $\eta = 0.4$ |
|---|---|---|---|---|---|
| CE[‡] | $71.33 \pm 0.43$ | $39.92 \pm 0.10$ | $7.59 \pm 0.20$ | $40.17 \pm 1.31$ | / |
| GCE[‡] | $63.09 \pm 1.39$ | $56.11 \pm 1.35$ | $17.42 \pm 0.06$ | $40.91 \pm 0.57$ | / |
| NCE+AGCE[‡] | $69.03 \pm 0.37$ | $59.47 \pm 0.36$ | $24.72 \pm 0.60$ | $43.76 \pm 0.70$ | / |
| CE | $\mathbf{77.44 \pm 0.13}$ | $49.96 \pm 0.02$ | $11.23 \pm 0.45$ | $45.73 \pm 0.49$ | $54.40 \pm 0.35$ |
| GCE | $73.88 \pm 0.25$ | $64.67 \pm 0.49$ | $23.90 \pm 2.69$ | $45.14 \pm 0.13$ | $56.95 \pm 0.63$ |
| NCE+AGCE | $76.37 \pm 0.25$ | $64.55 \pm 0.46$ | $\mathbf{26.19 \pm 1.14}$ | $40.93 \pm 1.22$ | $53.67 \pm 0.18$ |
| TCE | $58.04 \pm 1.15$ | $45.91 \pm 1.25$ | $20.47 \pm 1.45$ | $28.35 \pm 0.74$ | $32.22 \pm 1.22$ |
| TCE shift, $\tau = 4.2$ | $77.14 \pm 0.11$ | $60.17 \pm 0.47$ | $18.16 \pm 0.37$ | $44.56 \pm 0.71$ | $55.07 \pm 0.19$ |
| TCE scale, $\tau = 4.2$ | $75.79 \pm 0.17$ | $62.88 \pm 0.59$ | $20.78 \pm 1.53$ | $43.57 \pm 1.19$ | $56.13 \pm 0.22$ |
| MAE | $7.46 \pm 1.92$ | $4.65 \pm 1.55$ | $3.21 \pm 0.57$ | $1.61 \pm 0.53$ | $1.54 \pm 0.47$ |
| IMAE, $T = 10$ | $5.26 \pm 3.19$ | $45.01 \pm 1.10$ | $4.51 \pm 0.69$ | $48.09 \pm 0.68$ | $2.85 \pm 1.05$ |
| MAE shift, $\tau = 3.4$ | $76.65 \pm 0.30$ | $61.29 \pm 0.49$ | $19.30 \pm 1.00$ | $44.06 \pm 1.23$ | $54.83 \pm 0.49$ |
| MAE scale, $\tau = 3.4$ | $73.54 \pm 0.32$ | $\mathbf{64.92 \pm 0.20}$ | $23.00 \pm 2.44$ | $\mathbf{48.88 \pm 0.79}$ | $\mathbf{57.56 \pm 0.41}$ |

Table 5: Shifting or scaling $w(\boldsymbol{s}, y)$ mitigates underfitting on CIFAR100 under different label noise. We report test accuracies with 3 different runs. Both $\tau$ of the shifted/scaled MAE/TCE and $T$ of IMAE are tuned with symmetric noise $\eta = 0.4$. Previous best results from Zhou et al. (2021) are included as context (denoted with ‡). See Appendix B for results with more noise rates.

| Settings | CE | MAE | IMAE | MAE shift | MAE scale | TCE | TCE shift | TCE scale |
|---|---|---|---|---|---|---|---|---|
| $k = 50$ | 66.40 | 3.68 | | 60.76 | 66.72 | | | |
| $k = 200$ | 70.26 | 0.50 | | 59.31 | 71.92 | | | |
| $k = 400$ | 70.16 | 0.25 | | 47.32 | 71.87 | | | |

Table 6: Shifting or scaling $w(\boldsymbol{s}, y)$ mitigates underfitting on WebVision subsampled with different numbers of classes. $k = 50$ is the standard "mini" setting in previous work (Ma et al., 2020; Zhou et al., 2021). Hyperparameters roughly tuned for each setting are left to Table 10 of Appendix B. We report test accuracy with a single run due to a limited computation budget.

or shifting

$$w^+(\boldsymbol{s}, y) = w^+(\Delta(\boldsymbol{s}, y)) = w(\Delta(\boldsymbol{s}, y) + |\mathbb{E}_k[\Delta(\boldsymbol{s}, y)]| - \tau)$$

the sample-weighting functions, where $\tau$ is a hyperparameter. Intuitively, these approaches cancel the effect of large $k$ on the weight of $\mathbb{E}_k[\Delta(\boldsymbol{s}, y)]$ at initialization. Small $\tau$ thus lead to high initial sample weights regardless of $k$. Alternatively, Wang et al. (2019a) propose IMAE as *a fix to MAE*, which essentially applies a scaled exponential transform to $w_{\mathrm{MAE}}(\boldsymbol{s}, y)$

$$w_{\mathrm{IMAE}}(\boldsymbol{s}, y) = e^{T p_y (1 - p_y)}$$

where $T \geq 0$ is a hyperparameter. We visualize scaled and shifted sample-weighting functions of MAE and $w_{\mathrm{IMAE}}$[2] in Fig. 4c. Notably, $\lim_{\Delta(\boldsymbol{s}, y) \to \infty} w_{\mathrm{IMAE}}(\boldsymbol{s}, y) > 0$, which can lead to excessive learning of well-learned samples and thus risk overfitting. We report results on CIFAR100 with different label noise in Table 5, and results on the noisy large-scale dataset WebVision in Table 6. In summary, both shifting and scaling significantly improve performance of MAE, thus alleviating underfitting, making MAE comparable to the previous state-of-the-art.

## 5 CONCLUSION

We unify a broad array of loss functions into the same standard form, explicitly connecting the design of loss functions to the design of sample-weighting curricula. Based on the curriculum view, we gain more insights into how different design choices affect the training dynamics, especially the noise robustness and vulnerability to underfitting. Our theoretical and empirical findings can help design better loss functions and learning curricula in future work.

---

[2] $w_{\mathrm{IMAE}}$ is also normalized to unit maximum.

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

## A  REVIEW AND DERIVATIONS OF LOSS FUNCTIONS

We provide a comprehensive review of loss functions in Tables 1 and 2. We ignore the differences in constant scaling factors and additive bias as they can be absorb into other hyperparameters. We examine how hyperparameters affect different sample-weighting functions in Fig. 5.

## A.1 LOSS FUNCTIONS WITHOUT ROBUSTNESS GUARANTEES

We first review loss functions not satisfying Eq. (1).

**Cross Entropy** (CE)

$$L_{\mathrm{CE}}(\boldsymbol{s}, y) = -\log p_y$$

is the standard loss function for classification.

**Focal Loss** (FL; Lin et al. 2017)

$$L_{\mathrm{FL}}(\boldsymbol{s}, y) = -(1 - p_y)^q \log p_y$$

aims to address label imbalance when training object detection models.

**Symmetric Cross Entropy** (SCE; Wang et al. 2019c)

$$L_{\mathrm{SCE}}(\boldsymbol{s}, y) = a \cdot L_{\mathrm{CE}}(\boldsymbol{s}, y) + b \cdot L_{\mathrm{RCE}}(\boldsymbol{s}, y)$$
$$\propto (1 - q) \cdot (-\log p_y) + q \cdot (1 - p_y)$$

is a weighted average of CE and RCE (MAE), where $a > 0$, $b > 0$, and $0 < q < 1$. It exhibit better noise-robustness due to the combination with RCE (MAE).

## A.2 SYMMETRIC LOSS FUNCTIONS

A loss function $L$ is *symmetric* (Ghosh et al., 2017) if

$$\sum_{i=1}^{k} L(\boldsymbol{s}, i) = C, \ \forall \boldsymbol{s} \in \mathbb{R}^k,$$

with a constant $C$. It is proved to be robust against *symmetric* label noise when $\eta < (k-1)/k$.

**Mean Absolute Error** (MAE; Ghosh et al. 2017)

$$L_{\mathrm{MAE}}(\boldsymbol{s}, y) = \sum_{i=1}^{k} |\mathbb{I}(i = y) - p_i| = 2 - 2p_y \propto 1 - p_y$$

is a classic symmetric loss function, where $\mathbb{I}(i = y)$ is the indicator function.

**Reverse Cross Entropy** (RCE; Wang et al. 2019c)

$$L_{\mathrm{RCE}}(\boldsymbol{s}, y) = -\sum_{i=1}^{k} p_i \log \mathbb{I}(i = y) = -\sum_{i \neq y} p_i \log 0 = -(1 - p_y)A \propto 1 - p_y = L_{\mathrm{MAE}}(\boldsymbol{s}, y)$$

is equivalent to MAE in implementation, where $\mathbb{I}(\cdot)$ is the indicator function and $\log 0$ is truncated to a negative constant $A$ to avoid numerical overflow.

Ma et al. (2020) argued that any generic loss functions with $L(\boldsymbol{s}, i) > 0, \forall i \in \{1, ..., k\}$ can become symmetric by simply normalizing them. As an example,

**Normalized Cross Entropy** (NCE; Ma et al. 2020)

$$L_{\mathrm{NCE}}(\boldsymbol{s}, y) = \frac{L_{\mathrm{CE}}(\boldsymbol{s}, y)}{\sum_{i=1}^{k} L_{\mathrm{CE}}(\boldsymbol{s}, i)} = \frac{-\log p_y}{\sum_{i=1}^{k} -\log p_i}$$

is a symmetric loss function.

## A.3 ASYMMETRIC LOSS FUNCTIONS

$L$ as a function of the softmax probability $p_i$, $L(\boldsymbol{s}, i) = l(p_i)$, is *asymmetric* (Zhou et al., 2021) if

$$\tilde{r} = \max_{i \neq y} \frac{P(\tilde{y} = i | \boldsymbol{x}, y)}{P(\tilde{y} = y | \boldsymbol{x}, y)} \leq \inf_{\substack{0 \leq p_i, p_j \leq 1 \\ p_i + p_j \leq 1}} \frac{l(p_i) - l(p_i + p_j)}{l(0) - l(p_j)},$$

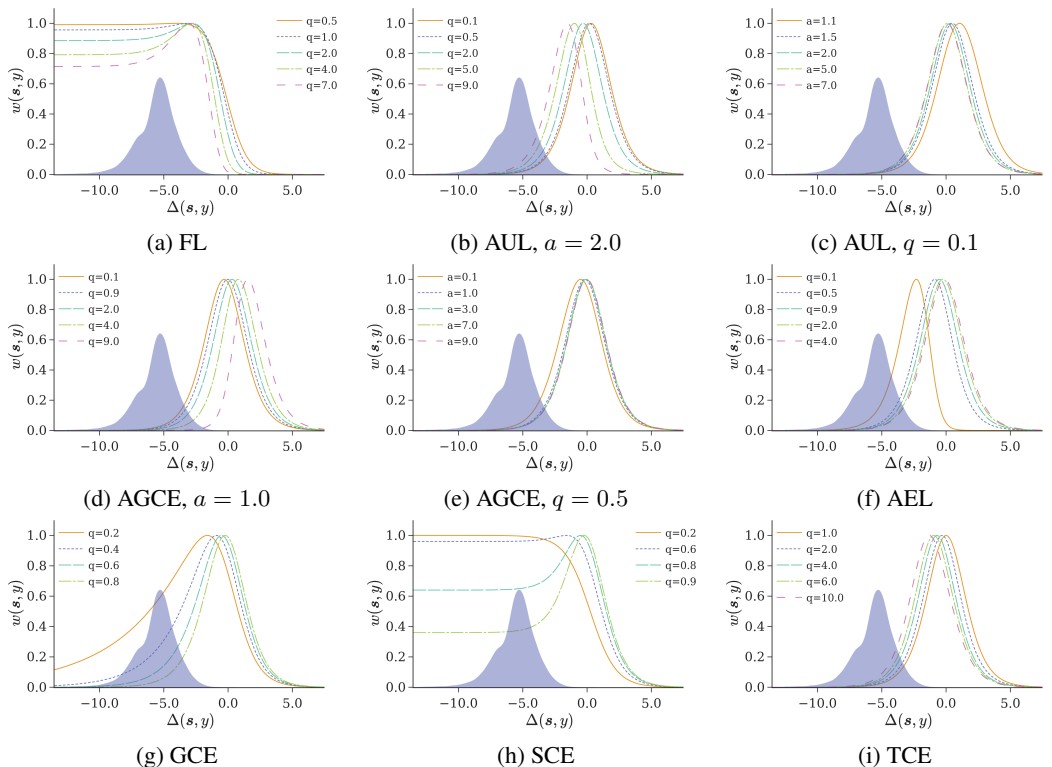

Figure 5: How hyperparameters affect the sample-weighting functions in Table 1. The initial $\Delta(\boldsymbol{s}, y)$ distributions of CIFAR100 extracted with a randomly initialized model are included as reference.

where $p_j$ is a valid increment of $p_i$. An asymmetric loss function is robust against *generic* label noise when $\tilde{r} < 1$, i.e., there are more clean samples than corrupted samples. Zhou et al. (2021) propose the following asymmetric loss functions with conditions to satisfy the asymmetric criterion:

**Asymmetric Generalized Cross Entropy** (AGCE)

$$L_{\text{AGCE}}(\boldsymbol{s}, y) = \frac{(a+1) - (a+p_y)^q}{q}$$

where $a > 0$ and $q > 0$. It is asymmetric when $\mathbb{I}(q \leq 1)(\frac{a+1}{a})^{1-q} + \mathbb{I}(q > 1) \leq 1/\tilde{r}$.

**Asymmetric Unhinged Loss** (AUL)

$$L_{\text{AUL}}(\boldsymbol{s}, y) = \frac{(a - p_y)^q - (a-1)^q}{q}$$

where $a > 1$ and $q > 0$. It is asymmetric when $\mathbb{I}(q \leq 1)(\frac{a}{a-1})^{q-1} + \mathbb{I}(q \leq 1) \leq 1/\tilde{r}$.

**Asymmetric Exponential Loss** (AEL)

$$L_{\text{AEL}}(\boldsymbol{s}, y) = e^{-p_y/q}$$

where $q > 0$. It is asymmetric when $e^{1/q} \leq 1/\tilde{r}$.

### A.4 OTHER ROBUST LOSS FUNCTIONS

**Generalized Cross Entropy** (GCE; Zhang & Sabuncu 2018)

$$L_{\text{GCE}}(\boldsymbol{s}, y) = \frac{1 - p_y^q}{q}$$

can be viewed as a smooth interpolation between CE and MAE, where $0 < q \leq 1$. CE or MAE can be recovered by setting $q \to 0$ or $q = 1$.

**Taylor Cross Entropy** (TCE; Feng et al. 2020)

$$L_{\mathrm{TCE}}(\boldsymbol{s}, y) = \sum_{i=1}^{q} \frac{(1 - p_y)^i}{i}$$

is derived from Taylor series of the $\log$ function in CE. It reduces to MAE when $q = 1$. Interestingly, the summand of TCE $(1 - p_y)^i / i$ with $i > 2$ is proportional to AUL with $a = 1$ and $q = i$. Thus TCE can be viewed as a combination of symmetric and asymmetric loss functions.

**Active-Passive Loss** (APL; Ma et al. 2020) Ma et al. (2020) propose weighted combinations of active and passive loss functions. Active loss follows $\forall i \neq y$, $l(\boldsymbol{s}, i) = 0$, which emphasize the learning of labeled class. Passive loss complies with $\exists i \neq y$, $l(\boldsymbol{s}, i) \neq 0$, which focus on unlearning other classes. In their derivation, NCE is active while MAE is passive. However, ambiguities arise when determine whether $l(\boldsymbol{s}, i)$ is active or pasive: given

$$L_{\mathrm{MAE}}(\boldsymbol{s}, y) \propto \sum_i |\mathbb{I}(i = y) - p_i| \propto \sum_i \mathbb{I}(i = y)(1 - p_i)$$

with $\mathbb{I}(\cdot)$ the indicator function, MAE can be active with $l(\boldsymbol{s}, i) = \mathbb{I}(i = y)(1 - p_i)$ but passive with $l(\boldsymbol{s}, i) = |\mathbb{I}(i = y) - p_i|$. We include NCE+MAE as an example:

$$L_{\mathrm{NCE+MAE}}(\boldsymbol{s}, y) = a \cdot L_{\mathrm{NCE}}(\boldsymbol{s}, y) + b \cdot L_{\mathrm{MAE}}(\boldsymbol{s}, y)$$
$$\propto (1 - q) \cdot \frac{-\log p_y}{\sum_{i=1}^{k} - \log p_i} + q \cdot (1 - p_y)$$

where $a > 0$, $b > 0$, and $0 < q < 1$.

## A.5 Loss Functions with Output Regularizers

We extract the output regularizers from existing loss functions.

**Mean Square Error** (MSE; Ghosh et al. 2017)

$$L_{\mathrm{MSE}}(\boldsymbol{s}, y) = \sum_{i=1}^{k} (\mathbb{I}(i = y) - p_i)^2 = 1 - 2p_y + \sum_{i=1}^{k} p_i^2$$
$$\propto 1 - p_y + \frac{1}{2} \cdot \sum_{i=1}^{k} p_i^2 = L_{\mathrm{MAE}}(\boldsymbol{s}, y) + \alpha \cdot R_{\mathrm{MSE}}(\boldsymbol{s})$$

is more robust than CE (Ghosh et al., 2017), where $\alpha = 0.5$ and the regularizer

$$R_{\mathrm{MSE}}(\boldsymbol{s}) = \sum_{i=1}^{k} p_i^2 \tag{6}$$

increases the entropy of the softmax output. We can generalize $\alpha$ to a hyperparamter, making MSE a combination of MAE and an entropy regularizer $R_{\mathrm{MSE}}$.

**Peer Loss** (PL; Liu & Guo 2020)

$$L_{\mathrm{PL}}(\boldsymbol{s}, y) = L(\boldsymbol{s}, y) - L(\mathbf{s}, \mathbf{y})$$

makes a generic loss function $L(\boldsymbol{s}, y)$ robust against label noise, where $\mathbf{s}$ denotes the score of an input $\mathbf{x}$ and $\mathbf{y}$ the label, both randomly sampled from the noisy data distribution $\mathcal{D}$. Its noise robustness is theoretically established for binary classification and extended to multi-class setting (Liu & Guo, 2020).

**Confidence Regularizer** (CR; Cheng et al. 2021)

$$R_{\mathrm{CR}}(\boldsymbol{s}) = -\mathbb{E}_{\mathbf{y} \sim \mathcal{D}, \mathbf{x} \sim \mathcal{D}}[\log p_{\mathbf{y}|\mathbf{x}}]$$

is shown (Cheng et al., 2021) to be the regularizer induced by PL in expectation. Minimizing $R_{\mathrm{CR}}(\boldsymbol{s})$ makes the softmax output distribution $\boldsymbol{p}$ deviate from the prior label distribution of the noisy dataset $P(\tilde{y} = i)$, reducing the entropy of the softmax output.

| Loss | CIFAR10 | | CIFAR100 | |
|---|---|---|---|---|
| | $a$ | $q$ | $a$ | $q$ |
| FL | / | 3.0 | / | 3.0 |
| AGCE | 5.4 | 1.5 | 0.1 | 0.1 |
| AUL | 6.1 | 4.8 | 2.0 | 8.7 |
| AEL | / | 5.0 | / | 0.2 |
| GCE | / | 0.9 | / | 0.7 |
| TCE | / | 2 | / | 6 |
| SCE | / | 0.9 | / | 0.15 |

Table 7: Hyperparameters of loss functions for results in Table 3 and Table 4, tuned with symmetric label noise $\eta = 0.4$ on CIFAR10 and CIFAR100, respectively.

**Generalized Label Smoothing** (GLS; Wei et al. 2021)

Lukasik et al. (2020) show that label smoothing (LS; Szegedy et al. 2016) can mitigate overfitting with label noise, which is later extended to GLS. Cross entropy with GLS is

$$L_{\text{CE+GLS}}(\boldsymbol{s}, y) = \sum_{i=1}^{k} -[\mathbb{I}(i = y)(1 - \alpha) + \frac{\alpha}{k}] \log p_i$$

$$= -(1 - \alpha) \log p_y - \alpha \cdot \frac{1}{k} \sum_{i=1}^{k} \log p_i$$

$$\propto -\log p_y - \frac{\alpha}{1 - \alpha} \cdot \frac{1}{k} \sum_{i=1}^{k} \log p_i = L_{\text{CE}}(\boldsymbol{s}, y) + \alpha' \cdot R_{\text{GLS}}(\boldsymbol{s})$$

where $\alpha' = \alpha/(1 - \alpha)$, has regularizer $R_{\text{GLS}}$

$$R_{\text{GLS}}(\boldsymbol{s}) = -\sum_{i=1}^{k} \frac{1}{k} \log p_i \tag{7}$$

With $\alpha' > 0$, $R_{\text{GLS}}$ corresponds to the original LS, which increases the entropy of softmax outputs. In contrast, $\alpha' < 0$ corresponding to Negative Label Smoothing (NLS; Wei et al. 2021), which decreases the output entropy similar to $R_{\text{CR}}$.

**Normalized Cross Entropy** (NCE; Ma et al. 2020)

$$L_{\text{NCE}}(\boldsymbol{s}, y) = \frac{L_{\text{CE}}(\boldsymbol{s}, y)}{\sum_{i=1}^{k} L_{\text{CE}}(\boldsymbol{s}, i)} = \frac{-\log p_y}{\sum_{i=1}^{k} -\log p_i}$$

has been reviewed in Appendix A.2. However, our derivations in §3 shows that it encompass an output regularizer

$$R_{\text{NCE}}(\boldsymbol{s}) = \sum_{i=1}^{k} \frac{1}{k} \log p_i$$

which is equivalent to NLS. **Jensen-Shannon Divergence** (JS; Englesson & Azizpour, 2021)

$$L_{\text{JS}}(\boldsymbol{s}, y) = a \sum_i p_i \log \frac{p_i}{ap_i + (1 - a)e_i} + (1 - a) \sum_i e_i \log \frac{e_i}{ap_i + (1 - a)e_i}$$

$$= a \sum_i p_i \log \frac{p_i}{ap_i + (1 - a)e_i} + (1 - a) \log \frac{1}{ap_y + (1 - a)}$$

$$= a \sum_{i \neq y} p_i \log \frac{p_i}{ap_i} + ap_y \log \frac{p_y}{ap_y + 1 - a} + (1 - a) \log \frac{1}{ap_y + (1 - a)}$$

$$= a \sum_i p_i \log \frac{p_i}{ap_i} - ap_y \log \frac{p_y}{ap_y} + ap_y \log \frac{p_y}{ap_y + 1 - a} + (1 - a) \log \frac{1}{ap_y + (1 - a)}$$

$$= ap_y \log ap_y - (ap_y + 1 - a) \log(ap_y + 1 - a) - a \log a \sum_i p_i$$

$$= L'_{\text{JS}}(\boldsymbol{s}, y) + a' R_{\text{JS}}(\boldsymbol{s})$$

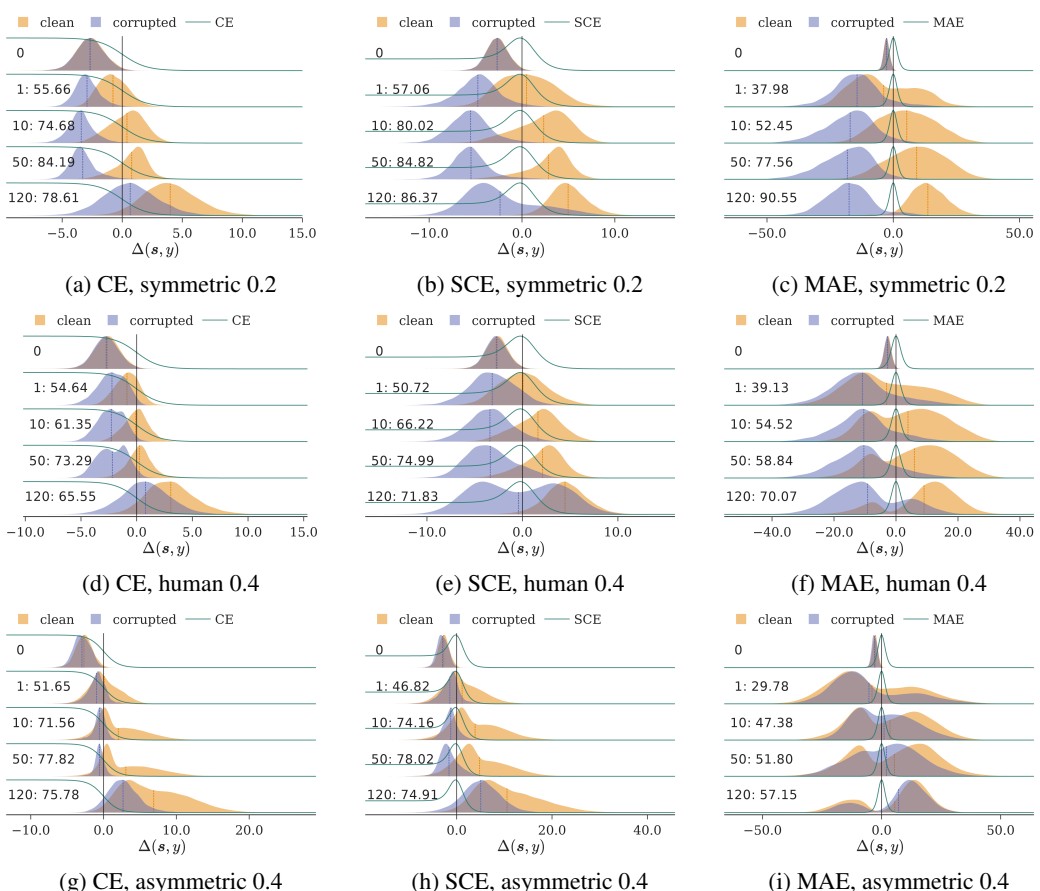

Figure 6: Additional results to Fig. 1 with different label noise and loss functions: (a-c) symmetric label noise with $\eta = 0.2$; (d-f) human label noise with $\eta = 0.4$; (g-i) asymmetric label noise with $\eta = 0.4$. We include the median line of each distribution, test accuracies and the sampled epochs for reference.

is proposed as another combination of CE and MAE, where $0 < a < 1$ is the hyperparameter with $\lim_{a \to 0} L_{\mathrm{JS}}(\boldsymbol{s}, y) = L_{\mathrm{CE}}(\boldsymbol{s}, y)$ and $\lim_{a \to 1} L_{\mathrm{JS}}(\boldsymbol{s}, y) = L_{\mathrm{MAE}}(\boldsymbol{s}, y)$. $a' = -a \log a$ is the weight for regularizer

$$R_{\mathrm{JS}}(\boldsymbol{s}) = \sum_i p_i$$

which can be regarded as a hyperparameter.

$$L'_{\mathrm{JS}}(\boldsymbol{s}, y) = a p_y \log a p_y - (a p_y + 1 - a) \log(a p_y + 1 - a)$$

is the primary loss function of JS conforming to Eq. (3) with

$$w_{\mathrm{JS}}(\boldsymbol{s}, y) = a p_y (p_y - 1)(\log a p_y - \log(a p_y + 1 - q))$$

## B  ADDITIONAL RESULTS OF THE TRAINING DYNAMICS

We complement §4 in the main text with detailed derivations, experiment settings and additional results. Hyperparameters of robust loss functions are summarized in Table 7.

**Additional results of Fig. 1.** In Fig. 6 and 7 we show extended results on the training dynamics with different label noise and loss functions. They follow the same trend as the results in Fig. 1 in the main text.

**Additional results of Fig. 2.** In Fig. 8 we show extended results on how different regularizers affect noise robust training. They follow the same trend as the results in Fig. 2 in the main text.

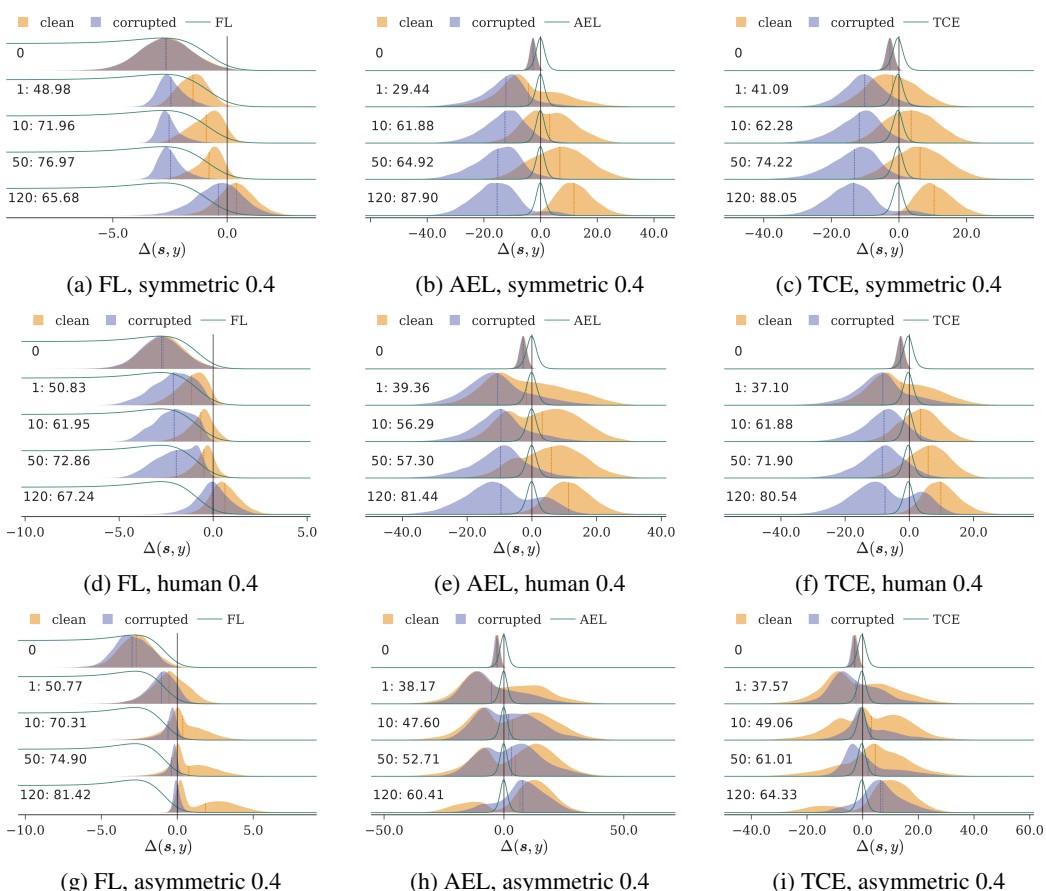

Figure 7: Additional results to Fig. 1 with different label noise and loss functions: (a-c) symmetric label noise with $\eta = 0.2$; (d-f) human label noise with $\eta = 0.4$; (g-i) asymmetric label noise with $\eta = 0.4$. We include the median line of each distribution, test accuracies and the sampled epochs for reference.

**Derivation of $\mathbb{E}_k[\Delta(s, y)]$ in Eq. (5)** Assume that class scores at *initialization* are i.i.d. normal variables $s_i \sim \mathcal{N}(\mu, \sigma)$,

$$\mathbb{E}_k[\Delta(s, y)] = \mathbb{E}[s_y - \log \sum_{i \neq y} e^{s_i}] = \mu - \mathbb{E}[\log \sum_{i \neq y} e^{s_i}]$$

$$\approx_1 \mu - \log \mathbb{E}[\sum_{i \neq y} e^{s_i}] + \frac{\mathbb{V}[\sum_{i \neq y} e^{s_i}]}{2\mathbb{E}[\sum_{i \neq y} e^{s_i}]^2}$$

$$=_2 \mu - \log\{(k-1)\mathbb{E}[e^{s_y}]\} + \frac{(k-1)\mathbb{V}[e^{s_y}]}{2\{(k-1)\mathbb{E}[e^{s_y}]\}^2}$$

$$=_3 \mu - \log[(k-1)e^{\mu+\sigma^2/2}] + \frac{(k-1)(e^{\sigma^2}-1)e^{2\mu+\sigma^2}}{2[(k-1)e^{\mu+\sigma^2/2}]^2}$$

$$= -\log(k-1) - \sigma^2/2 + \frac{e^{\sigma^2}-1}{2(k-1)}$$

where $\approx_1$ follows the approximation with Taylor expansion $\mathbb{E}[\log X] \approx \log \mathbb{E}[X] - \mathbb{V}[X]/(2\mathbb{E}[X]^2)$ (Teh et al., 2006), $=_2$ utilizes properties of sum of log-normal variables (Cobb et al., 2012), and $=_3$ substitutes $\mathbb{E}[e^{s_y}]$ and $\mathbb{V}[e^{s_y}]$ with expressions for log-normal distributions.

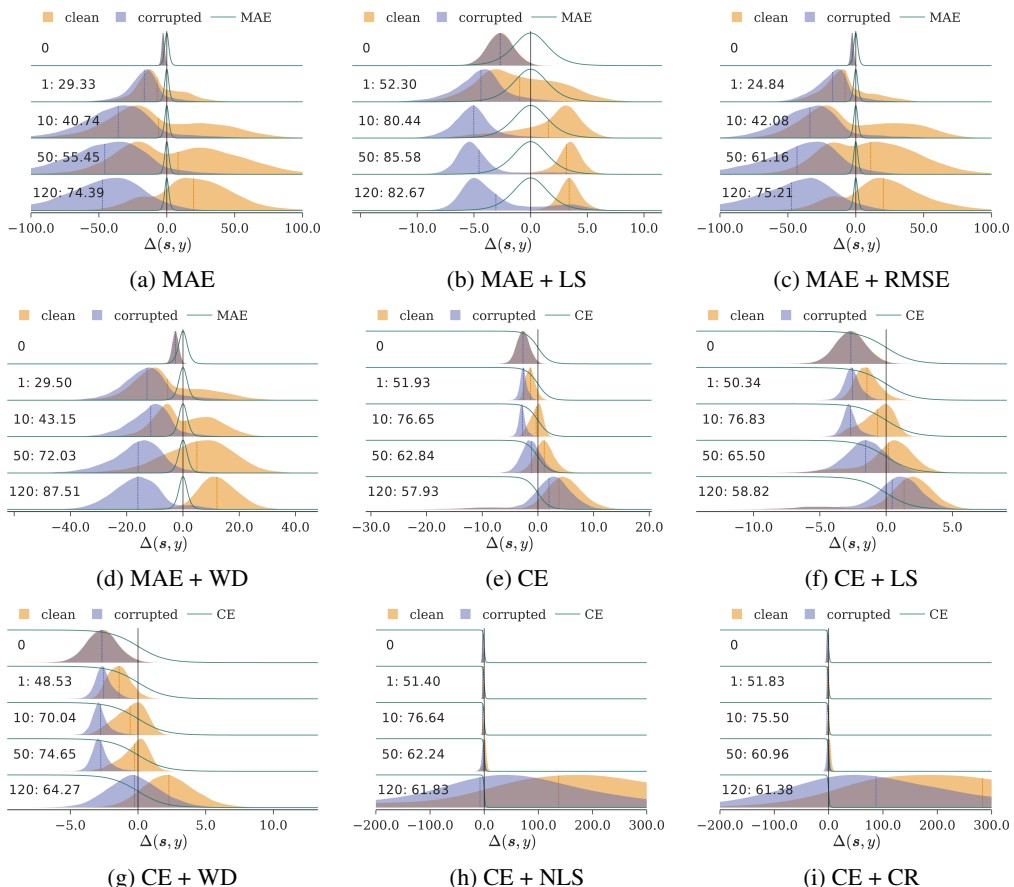

Figure 8: Training dynamics with various loss functions and regularizers on CIFAR10 with symmetric noise $\eta = 0.4$. We include the median (dashed) of each distribution and $-\log k$ (solid) lines and the test accuracy of each epoch.

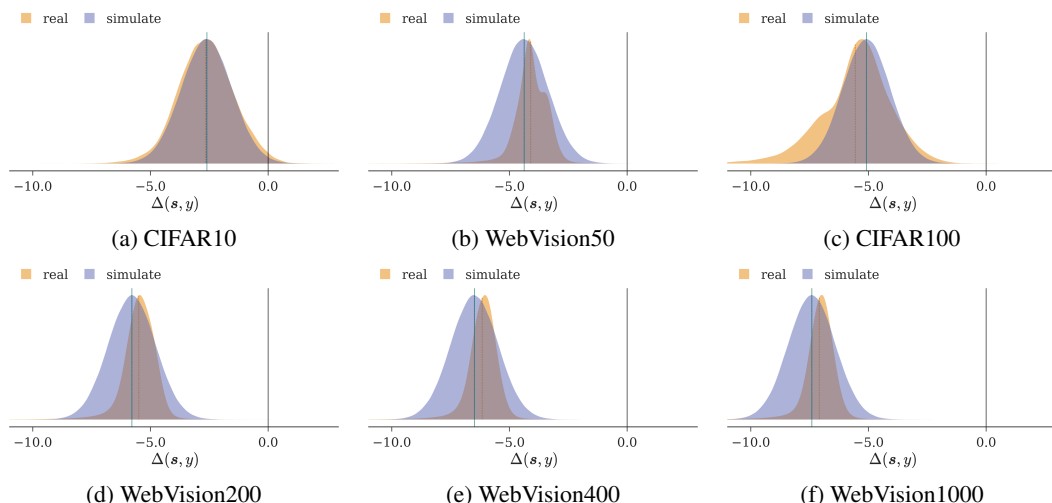

Figure 9: Comparisons between shapes of the simulated and real $\Delta(s, y)$ distributions at initialization. Simulations are based on the assumption that class scores follow normal distribution $s_i \sim \mathcal{N}(0, 1)$ at initialization. Real distributions are extracted with randomly initialized models. We also mark the estimated $\mathbb{E}_k[\Delta(s, y)]$ with vertical green lines.

| | Clean | Symmetric | | | |
|---|---|---|---|---|---|
| Loss | $\eta = 0$ | $\eta = 0.2$ | $\eta = 0.4$ | $\eta = 0.6$ | $\eta = 0.8$ |
| CE[‡] | $71.33 \pm 0.43$ | $56.51 \pm 0.39$ | $39.92 \pm 0.10$ | $21.39 \pm 1.17$ | $7.59 \pm 0.20$ |
| GCE[‡] | $63.09 \pm 1.39$ | $61.57 \pm 1.06$ | $56.11 \pm 1.35$ | $45.28 \pm 0.61$ | $17.42 \pm 0.06$ |
| NCE+TCE[‡] | $69.03 \pm 0.37$ | $65.66 \pm 0.46$ | $59.47 \pm 0.36$ | $48.02 \pm 0.58$ | $24.72 \pm 0.60$ |
| CE | $\mathbf{77.44 \pm 0.13}$ | $64.84 \pm 0.19$ | $49.96 \pm 0.02$ | $30.41 \pm 0.60$ | $11.23 \pm 0.45$ |
| GCE | $73.88 \pm 0.25$ | $70.55 \pm 0.73$ | $64.67 \pm 0.49$ | $52.57 \pm 0.47$ | $23.90 \pm 2.69$ |
| NCE+AGCE | $76.37 \pm 0.25$ | $\mathbf{72.19 \pm 0.26}$ | $64.55 \pm 0.46$ | $52.79 \pm 0.22$ | $\mathbf{26.19 \pm 1.14}$ |
| TCE | $58.04 \pm 1.15$ | $52.13 \pm 1.92$ | $45.91 \pm 1.25$ | $33.64 \pm 0.22$ | $20.47 \pm 1.45$ |
| TCE shift, $\tau = 4.2$ | $77.14 \pm 0.11$ | $71.28 \pm 0.44$ | $60.17 \pm 0.47$ | $40.91 \pm 0.33$ | $18.16 \pm 0.37$ |
| TCE scale, $\tau = 4.2$ | $75.79 \pm 0.17$ | $71.11 \pm 0.39$ | $62.88 \pm 0.59$ | $46.52 \pm 0.39$ | $20.78 \pm 1.53$ |
| MAE | $7.46 \pm 1.92$ | $4.65 \pm 1.55$ | $3.21 \pm 0.57$ | $1.61 \pm 0.53$ | $1.54 \pm 0.47$ |
| IMAE, $T = 10$ | $5.26 \pm 3.19$ | $11.81 \pm 3.10$ | $45.01 \pm 1.10$ | $28.94 \pm 1.92$ | $4.51 \pm 0.69$ |
| MAE shift | $76.65 \pm 0.30$ | $71.36 \pm 0.70$ | $61.29 \pm 0.49$ | $42.68 \pm 1.14$ | $19.30 \pm 1.00$ |
| MAE scale | $73.54 \pm 0.32$ | $69.96 \pm 0.55$ | $\mathbf{64.92 \pm 0.20}$ | $\mathbf{54.98 \pm 0.17}$ | $23.00 \pm 2.44$ |

Table 8: Extended result of Table 5: shifting or scaling $w(s, y)$ mitigates underfitting on CIFAR100 with symmetric label noise. We report test accuracies with 3 different runs. Both $\tau$ of the shifted/scaled MAE/TCE and $T$ of IMAE are tuned with symmetric noise $\eta = 0.4$. Previous best results from Zhou et al. (2021) are included as context (denoted with ‡). Previous best results from Zhou et al. (2021) are included as context (denoted with ‡).

**Simulated $\Delta(s, y)$ approximate real settings.** We compare the simulated $\Delta(s, y)$ distributions based on $s_i \sim \mathcal{N}(0, 1)$ to distributions of real datasets at initialization in Fig. 9. The expectations of simulated $\Delta(s, y)$ are similar to real settings, which supports the analysis in §4.3. The estimated $\mathbb{E}_k[\Delta(s, y)]$ fits the median of simulation well. Finally, increasing the number of classes $k$ consistently decreases $\Delta(s, y)$ at initialization.

**Additional results on the effectiveness of shifting and scaling $w(s, y)$.** We show additional results on how shifting and scaling $w(s, y)$ affect noise robust training in Tables 8 and 9. Hyperparameters for results of the WebVision dataset are shown in

| Loss | Clean $\eta = 0$ | Asymmetric $\eta = 0.1$ | $\eta = 0.2$ | $\eta = 0.3$ | $\eta = 0.4$ |
|---|---|---|---|---|---|
| CE[‡] | $71.33 \pm 0.43$ | $64.85 \pm 0.37$ | $58.11 \pm 0.32$ | $50.68 \pm 0.55$ | $40.17 \pm 1.31$ |
| GCE[‡] | $63.09 \pm 1.39$ | $63.01 \pm 1.01$ | $59.35 \pm 1.10$ | $53.83 \pm 0.64$ | $40.91 \pm 0.57$ |
| NCE+AGCE[‡] | $69.03 \pm 0.37$ | $67.22 \pm 0.12$ | $63.69 \pm 0.19$ | $55.93 \pm 0.38$ | $43.76 \pm 0.70$ |
| CE | $\mathbf{77.44 \pm 0.13}$ | $72.08 \pm 0.19$ | $64.75 \pm 0.49$ | $55.62 \pm 0.12$ | $45.73 \pm 0.49$ |
| GCE | $73.88 \pm 0.25$ | $72.29 \pm 0.16$ | $67.96 \pm 0.24$ | $57.83 \pm 1.02$ | $45.14 \pm 0.13$ |
| NCE+AGCE | $76.37 \pm 0.25$ | $\mathbf{73.73 \pm 0.21}$ | $64.58 \pm 0.37$ | $52.23 \pm 1.18$ | $40.93 \pm 1.22$ |
| TCE | $58.04 \pm 1.15$ | $53.51 \pm 1.18$ | $45.14 \pm 1.25$ | $34.59 \pm 2.47$ | $28.35 \pm 0.74$ |
| TCE shift, $\tau = 4.2$ | $77.14 \pm 0.11$ | $71.82 \pm 0.60$ | $63.41 \pm 0.06$ | $54.91 \pm 0.47$ | $44.56 \pm 0.71$ |
| TCE scale, $\tau = 4.2$ | $75.79 \pm 0.17$ | $72.64 \pm 0.48$ | $63.58 \pm 0.32$ | $53.73 \pm 0.58$ | $43.57 \pm 1.19$ |
| MAE | $7.46 \pm 1.92$ | $4.65 \pm 1.55$ | $3.21 \pm 0.57$ | $1.61 \pm 0.53$ | $1.54 \pm 0.47$ |
| IMAE, $T = 10$ | $5.26 \pm 3.19$ | $10.05 \pm 2.16$ | $14.90 \pm 1.41$ | $32.61 \pm 0.24$ | $48.09 \pm 0.68$ |
| MAE shift, $\tau = 3.4$ | $76.65 \pm 0.30$ | $71.82 \pm 0.44$ | $63.37 \pm 0.28$ | $53.81 \pm 0.57$ | $44.06 \pm 1.23$ |
| MAE scale, $\tau = 3.4$ | $73.54 \pm 0.32$ | $71.85 \pm 0.31$ | $\mathbf{69.43 \pm 0.39}$ | $\mathbf{63.14 \pm 0.37}$ | $\mathbf{48.88 \pm 0.79}$ |

Table 9: Extended result of Table 5: shifting or scaling $w(\boldsymbol{s}, y)$ mitigates underfitting on CI-FAR100 with asymmetric label noise. We report test accuracies with 3 different runs. Both $\tau$ of the shifted/scaled MAE/TCE and $T$ of IMAE are tuned with symmetric noise $\eta = 0.4$. Previous best results from Zhou et al. (2021) are included as context (denoted with ‡). Previous best results from Zhou et al. (2021) are included as context (denoted with ‡).

| Settings | CE | MAE | IMAE | MAE shift | MAE scale | TCE | TCE shift | TCE scale |
|---|---|---|---|---|---|---|---|---|
| $k = 50$ | / | / | $T =?$ | $\tau = 2.0$ | $\tau = 2.0$ | $q =?$ | $q =?, \tau =?$ | $q =?, \tau =?$ |
| $k = 200$ | / | / | $T =?$ | $\tau = 1.8$ | $\tau = 1.8$ | $q =?$ | $q =?, \tau =?$ | $q =?, \tau =?$ |
| $k = 400$ | / | / | $T =?$ | $\tau = 1.6$ | $\tau = 1.6$ | $q =?$ | $q =?, \tau =?$ | $q =?, \tau =?$ |

Table 10: Hyperparameters roughly tuned with each settings for results of WebVision in Table 6. For TCE shift/scale, we only tune $\tau$ and leave $q$ intact to demonstrate the effect of the shift/scale fixes.

