# OpenReview forum: "A Curriculum View of Robust Loss Functions"
_ICLR.cc/2024/Conference — Submitted to ICLR 2024_

### Official Review · Reviewer_gQaq · 2023-10-19

**Soundness:** 3 good
**Presentation:** 3 good
**Contribution:** 3 good
**Rating:** 6
**Confidence:** 4

**Summary:**

This paper proposes a new perspective for robust losses when learning with noisy labels. The authors show that most robust loss functions differ only in the sample-weighting curriculums they implicitly define with an optional implicit regularizer. This fills in the explanation of the dynamic performance of robust losses in training. Then the authors show the effects of loss functions and regularizers on learning through empirical studies, respectively.

**Strengths:**

- The motivation for understanding empirical phenomenons of robust losses against label noise is interesting. The common features different losses shown make the understanding significant. Good motivation, especially since the data are becoming larger and learning with noisy labels is becoming a pressing challenge.

- The empicial results echo the theoretical study. This paper conducts extensive experiments and comprehensive studies to evaluate the losses and regularizers, which helps give suggestions for using them.

- Experiments are well designed. The theoretical statements in this paper seem correct. I think this work is valuable.

**Weaknesses:**

- The robust losses involves the curriculum view, i.e., a sample-weighting perspective, the paper should include more discussion and empirical comparisons with curriculum and reweighting noisy-label learning methods.

- Eq.3 appears to rely on a number of assumptions, which should be clarified in the formulation of these assumptions.

- As one of the main formulas of the paper, it is not quite clear how Eq.4 was obtained. More details should be provided.

**Questions:**

The results and findings in this paper are insightful and would be useful for future research. The paper is also well-written with solid theoretical exposition and strong results. Overall, this is a good paper.

---

> ### Author Response · Authors · 2023-11-20
>
> Thanks for your comments!
>
> ### Regarding the Weaknesses
>
> > Q1: The robust losses involves the curriculum view, i.e., a sample-weighting perspective, the paper should include more discussion and empirical comparisons with curriculum and reweighting noisy-label learning methods.
>
> We have discussed the relation to existing curriculum approaches in the related work section (now the second paragraph in the revised version).
>
> We believe that empirical comparisons with existing sample-weighting curricula is optional, or even orthogonal to our experiments aiming to **understand existing loss functions**:
>
> - Results in section 4.1 analyze the effects of different implicit sample-weighting functions $w(\boldsymbol{s}, y)$ of loss functions **considered in this work**. We do no claim for a comprehensive analysis of all sample weighting curricula.
> - Results in section 4.2 analyze the effects of the identified regularizers within loss functions considered in this work. They are orthogonal to existing sample-weighting curricula.
> - Results in section 4.3 aim to identify the cause of underfitting. We first establish the relation between minimal sample weights to underfitting. We then theoretically connects minimal sample weights to the increased number of classes, which leads to minimal $\Delta(\boldsymbol{s}, y)$ and thus minimal weights at initialization with robust loss functions. We then test whether increasing the sample weights with small $\Delta(\boldsymbol{s}, y)$ helps mitigate underfitting:
>   - Better results with our fixes compared to the underfitting loss functions support our explanation.
>   - Results of previous state-of-the-art under the same experiment settings reflect the significance of the improvement  (we choose results from "Asymmetric loss functions for learning with noisy labels")
>   - We never claim for a "novel approach for new state-of-the-art performance", which does require comprehensive comparison with major approaches of noisy-robust learning.
>
> Given the page limitation, including additional empirical results of sample-weighting curriculums requires remove other important discussions of our unified standard form. We will reproduce some sample-weighting curricula in the appendix in later revisions.
>
> > Eq.3 appears to rely on a number of assumptions, which should be clarified in the formulation of these assumptions.
>
> We have thoroughly revised our derivations in Section 3 to clarify the assumptions and intuitions. Eq.3 $L'(\boldsymbol{s}, y) = - w(\boldsymbol{s}, y) \cdot \Delta(\boldsymbol{s}, y)$ is based on a mild assumption that we are using first order optimizers based on gradients.
>
> > As one of the main formulas of the paper, it is not quite clear how Eq.4 was obtained. More details should be provided.
>
> As mentioned above, the revised derivations in Section 3 makes Eq. 4 much clearer. We describe the example derivation of NCE towards Eq.4 before presenting the detailed standard form of Eq.4. Intuitively, we extract the regularizer $R(\boldsymbol{s})$ and rearrange other terms depending only on $\Delta(\boldsymbol{s}, y)$ into the primary loss function. If there is no explicit way to rearrange in the original loss terms, we turn to the gradients of $\boldsymbol{s}$, try rearranging the gradient terms, and then integrate over $\boldsymbol{s}$ after detaching weighting terms from derivative and integral. See the Appendix A.5 for detailed derivation for each loss functions.

---

### Official Review · Reviewer_pjXu · 2023-10-30

**Soundness:** 3 good
**Presentation:** 3 good
**Contribution:** 2 fair
**Rating:** 5
**Confidence:** 4

**Summary:**

This paper investigates robust loss functions used in learning with noisy labels. This paper unifies a broad array of loss functions into a novel standard form, which consists of a primary loss function inducing a sample-weighting curriculum and an optional implicit regularizer. The resulting curriculum view leads to a straightforward analysis of the training dynamics, which may help demystify how loss functions and regularizers affect learning and noise robustness. This paper shows that robust loss functions implicitly sift and neglect corrupted samples, and analyze the roles of regularizers with different loss functions. Finally, this paper proposes effective fixes to address the underfitting issue of robust loss functions.

**Strengths:**

- A novel curriculum perspective of robust loss functions is proposed, which consists of a primary loss function inducing a sample-weighting curriculum and an optional implicit regularizer.
- A sufficient number of robust loss functions are reviewed.
- The proposed simple fix seems to work well.

**Weaknesses:**

- Although a novel perspective that includes many loss functions is proposed, this perspective is unable to guide us to obtain any better robust loss functions, except for the simple fix to alleviate the underfitting issue.
- To me, the fix derived from the curriculum view is the central part of this paper. However, this paper did not provide extensive experiments to empirically validate this method. Is this method versatile enough? Can any robust loss functions (excluding MAE) be equipped with this method? Can this method work well on a variety of large-scale datasets? Without clearly and extensively demonstrating the effectiveness, the key contribution this paper seems limited.

**Questions:**

- Is the proposed fix versatile enough?
- Can any robust loss functions (excluding MAE) be equipped with this method?
- Can this method work well on a variety of large-scale datasets?

---

> ### Author Response · Authors · 2023-11-20
> **Response to Weakness 1**
>
> Thanks for your comments!
>
> ### Regarding the weaknesses
>
> > Q1: Although a novel perspective that includes many loss functions is proposed, this perspective is unable to guide us to obtain any better robust loss functions, except for the simple fix to alleviate the underfitting issue.
>
> It would be unfair to state that
>
> > this perspective is unable to guide us to obtain any better robust loss functions, except for the simple fix to alleviate the underfitting issue
>
> We should stress that:
>
> - Unifying loss functions **with different theoretical motivations** into the **same** dimensions of design choices
> - Connecting the design of loss functions to the design of learning curricula, which are commonly considered as distinct approaches.
>
> are theoretically **nontrivial**. These approaches are previously understood and analyzed with **distinct** approaches and motivations. Our theoretical derivations provide a **common** ground to understand and analyze them.
>
> As loss functions can be viewed as sample-weighting curricula, our findings suggest plenty of approaches to directly design them. As a quick list:
>
> 1. Blending the implcit curricula of robust loss functions to those explicitly designed, e.g., approach proposed in "DivideMix: Learning with Noisy Labels as Semi-supervised Learning"
> 2. We can use other metrics than $\Delta(\boldsymbol{s}, y)$ to identify the constantly unlearned samples, e.g., $\Delta(\boldsymbol{s}, y) - m$  where $m$ is a running average of $\Delta(\boldsymbol{s}, y)$, and assign small weights to small $\Delta(\boldsymbol{s}, y) - m$, which can avoid the underfitting issue due to minimal initial sample weights.
> 3. Use a proper regularizer for a good $\Delta(\boldsymbol{s}^*, y)$ with $\boldsymbol{s}^* = \arg\min_{\boldsymbol{s}} R(\boldsymbol{s})$ to facillitate learning with a large number of classes and avoid overfitting noisy samples. Existing confidence reducing regularizers all share the same minimum with $\Delta(\boldsymbol{s}^*, y) = - \log k$ where $k$ is the number of classes, which may risk underfitting with robust loss functions under large $k$.
> 4. Utilize the sample sifting effect of robust loss functions to **select** corrupted samples and either drop them or utilize semi-supervised learning to obtain surrogate labels.

---

> ### Author Response · Authors · 2023-11-20
> **Response to weakness 2 and questions**
>
> > Q2: To me, the fix derived from the curriculum view is the central part of this paper. However, this paper did not provide extensive experiments to empirically validate this method. Is this method versatile enough? Can any robust loss functions (excluding MAE) be equipped with this method? Can this method work well on a variety of large-scale datasets? Without clearly and extensively demonstrating the effectiveness, the key contribution this paper seems limited.
>
> It would be unfair to view the fixes for underfitting the central part of the paper, which ignores the significance of our theoretical framework and the following empirical insights:
>
> 1. Robust loss functions act as a sift to neglect corrupted samples. This finding provides a distinct view compared to the theoretical bounds of these loss functions.
> 2. Regularizers like weight decay is important to the learning of robust loss functions. It suggests that one should carefully tune the regularization with robust loss functions.
> 3. Underfitting is due to minimal sample weights at initilizations, which is empirically supported by our fixes.
>
> We have make these points more pronounced in the revised Introduction.
>
> We focus more on comphrehensively understand the loss functions considered in this work. The empirical performance of our fix aims to confirm whether avoiding minimal initial samples weights can improve the underfitting loss functions. The effectiveness on the underfitting MAE (and TCE in our revision) supports our explanation.
>
>
>
> As for the questions,
>
> > Is this method versatile enough?
>
> We don't understand what "versatile" means. Our fixes apply to underfitting with minimal intial sample weights. Nevertheless, our analysis of underfitting can help diagnose the cause in other settings. Notably, the novel curriculum view and the resulting analysis are more versatile than methods tailored for specific settings.
>
> > Can any robust loss functions (excluding MAE) be equipped with this method?
>
> Yes. As shown in Table 4, TCE also suffers to moderate underfitting. We include the results of our fixes to TCE on CIFAR100 in the revised Tables 5,8,9:
>
> |                         | Clean            | Symmmetric       | Symmetric        | Asymmetric       | Human            |
> | ----------------------- | ---------------- | ---------------- | ---------------- | ---------------- | ---------------- |
> |                         | $\eta=0$         | $\eta =0.4$      | $\eta =0.8$      | $\eta =0.4$      | $\eta =0.4$      |
> | CE                      | 77.44 $\pm$ 0.13 | 49.96 $\pm$ 0.02 | 11.23 $\pm$ 0.45 | 45.73 $\pm$ 0.49 | 54.40 $\pm$ 0.35 |
> | GCE                     | 73.88 $\pm$ 0.25 | 64.67 $\pm$ 0.49 | 23.90 $\pm$ 2.69 | 45.14 $\pm$ 0.13 | 56.95 $\pm$ 0.63 |
> | NCE+AGCE                | 76.37 $\pm$ 0.25 | 64.55 $\pm$ 0.46 | 26.19 $\pm$ 1.14 | 40.93 $\pm$ 1.22 | 53.67 $\pm$ 0.18 |
> | TCE                     | 58.04 $\pm$ 1.15 | 45.91 $\pm$ 1.25 | 20.47 $\pm$ 1.45 | 28.35 $\pm$ 0.74 | 32.22 $\pm$ 1.22 |
> | TCE shift, $\tau = 4.2$ | 77.14 $\pm$ 0.11 | 60.17 $\pm$ 0.47 | 18.16 $\pm$ 0.37 | 44.56 $\pm$ 0.71 | 55.07 $\pm$ 0.19 |
> | TCE scale, $\tau = 4.2$ | 75.79 $\pm$ 0.17 | 62.88 $\pm$ 0.59 | 20.78 $\pm$ 1.53 | 43.57 $\pm$ 1.19 | 56.13 $\pm$ 0.22 |
> | MAE                     | 7.46 $\pm$ 1.92  | 4.65 $\pm$ 1.55  | 3.21 $\pm$ 0.57  | 1.61 $\pm$ 0.53  | 1.54 $\pm$ 0.47  |
> | MAE shift, $\tau = 3.4$ | 76.65 $\pm$ 0.30 | 61.29 $\pm$ 0.49 | 19.30 $\pm$ 1.00 | 44.06 $\pm$ 1.23 | 54.83 $\pm$ 0.49 |
> | MAE scale, $\tau = 3.4$ | 73.54 $\pm$ 0.32 | 64.92 $\pm$ 0.20 | 23.00 $\pm$ 2.44 | 48.88 $\pm$ 0.79 | 57.56 $\pm$ 0.41 |
>
> Results on WebVision will be included in later revision as they requires weeks to converge.
>
> > Can this method work well on a variety of large-scale datasets?
>
> Yes. To our knowledge, WebVision is the largest noisy dataset of image classification. We have extend the standard mini setting (50 classes) to include more classes (200 and 400), which are much more challenging than settings in previous research.

---

### Official Review · Reviewer_L9A1 · 2023-11-03

**Soundness:** 3 good
**Presentation:** 2 fair
**Contribution:** 2 fair
**Rating:** 5
**Confidence:** 3

**Summary:**

The goal of the paper is to study the behaviour of robust loss functions during training from a sample reweighting perspective. To do this, the authors propose to rewrite each loss on a common form that has the same gradients as the original loss. This common form is of a gradient weight $w$ times a margin difference $\Delta$ (they also consider a more general form taking regularization into account). The authors then study these two quantities for different loss functions during training for clean and noisy labelled examples separately. For example, i) they find that gradient weights are higher for clean samples for the losses that generalize the best, ii) Mean Absolute Error (MAE) first learns easy examples which causes lower $\Delta$ of noisy examples (“sample sifting”), iii) label smoothing increases convergence and separations in $\Delta$. Finally, the authors study the underfitting issues of MAE via $w$ and $\Delta$ and proposes an effective fix.

Therefore, the contributions of the paper are to rewrite the losses on a common form, their observations, as well as the modification of MAE.

**Strengths:**

I find the paper has a well-organized structure which makes the high-level ideas easy to follow. I did not notice any technical issues in the paper. Furthermore, I like that mean and standard deviation were reported in Table 6, to show have some indication of variance between runs. The results are reproducible as the training setup (architecture, learning rate, weight decay, etc) are clearly stated including what hyperparameters were used and how they were selected. As far as I know, there is theoretical novelty in rewriting of the losses into a common form, and algorithmically in the proposed change to MAE.

**Weaknesses:**

I immensely value research papers improving our understanding and not just present a new method, which I believe is the goal of this paper. Having said that, I believe the significance of this work could be improved considerably by making the explanations, findings, and conclusions **clearer**:
* What is the motivation for rewriting the losses in this particular way? What novel perspective does this form give over something more intuitive like the $p_y$ and the gradient magnitude of the loss wrt to the logits? For example, in Figure 1, couldn’t one see the same thing with an x-axis with $p_y$ instead of $\Delta$, and the y-axis being the gradient magnitude wrt the logits for the MAE green curve instead of $w$?
* How isn’t it trivial that different loss functions have different gradient weights, and therefore different sample weights? If all loss functions penalized the same, there would be no reason to have different ones?
* A single sentence motivating the inequality in Equation 5 would make it clearer.
* The clarity of several equations could be improved:
  * The equation between Equations 2 and 3 is crucial and could be made much clearer: i) a single sentence discussion or motivation for using stop gradient, ii) what $\Delta_y$ is, iii) why two additional minus signs are added in the middle equation, iv) and as $w$ is a key component of the paper, I believe it deserves a proper definition.
  * The equation at the end of page 3: i) why are stop gradients used in that derivation? ii) why is a factor of k and 1/k introduced?
* It was unclear to me, why the more general form that accounts for regularization was introduced, when its properties like $R(s)$ never were studied.

**Experimental Rigor.**

Reporting mean and standard deviation would improve the conclusiveness of the observations in the tables. Furthermore, the network predictions from several runs could be used to have more reliable histogram estimates (more data).

**Novelty and Significance.**
* What novelty and significance does this work add over that of Wang et al. [1]?
  * What are the benefits of viewing the losses in terms of the gradient of $\Delta$ rather than the more natural gradient magnitude wrt to the logits?
  * The following finding is not novel: “Zhang & Sabuncu (2018) attribute underfitting of MAE to the lack of the 1/py term in sample gradients, which “treats every sample equally” and thus hampers learning. In contrast, we show that MAE emphasizes samples with moderate ∆(s, y).” That MAE does not treat samples equally and instead focus on examples with moderate loss/$\Delta$, is not novel in this work. This is one of the main findings of Wang et al., which is clearly shown in their Figure 1 (or 2 depending on version of the paper), where examples with low and high $p_y$ have small gradients.
  * I believe the following quote is misrepresenting the related work: “.. attribute underfitting to their low variance, making clean and noise samples less distinguishable. But as shown in Table 4 MAE can underfit data with clean labels.”. Wang et al. did not mention that the variance was low between clean and noisy samples, but rather the more general “informative” and “uninformative” examples. Furthermore, the finding that MAE underfits clean examples is not novel, and even Wang et al. clearly shows this in their Table 1.
  * Finally, Wang et al. already proposed a fix for MAE. Therefore, the novelty and significance of the proposed fix in this paper is unclear. A proper comparison (theoretically and experimentally) with other fixes for MAE is required.
* What’s the significance of the proposed way of rewriting the loss functions? If one instead does similar studies in terms of loss values (e.g., Figure 3 in [1], and Figure 2 in [2]) or gradients (e.g., Theorem 1 in [3]), it seems many of the findings in the paper are already well-known. For example, that the gradients of clean examples dominates the early learning phase and then the gradients for noisy labelled examples take over, resulting in overfitting [3]. Another example, that regularization methods If there are any novel and significant findings, I believe the authors should much more clearly state, relate, and discuss the significance of them compared to related work. That MAE focuses on moderate loss examples, as shown by Wang et al. (2019a).


**Missing related work.**

The list of robust loss functions is comprehensive, but missing some relevant ones based on information theory: i) f-divergences [5], ii) Bregman divergences [6], and iii) Jensen-Shannon divergences [7].

**References.**

[1] Wang X, Hua Y, Kodirov E, Robertson NM. Imae for noise-robust learning: Mean absolute error does not treat examples equally and gradient magnitude's variance matters.

[2] Chen P, Chen G, Ye J, Heng PA. Noise against noise: stochastic label noise helps combat inherent label noise.

[3] Li J, Socher R, Hoi SC. Dividemix: Learning with noisy labels as semi-supervised learning.

[4] Liu S, Niles-Weed J, Razavian N, Fernandez-Granda C. Early-learning regularization prevents memorization of noisy labels.

[5] Wei J, Liu Y. When optimizing $ f $-divergence is robust with label noise.

[6] Amid E, Warmuth MK, Anil R, Koren T. Robust bi-tempered logistic loss based on bregman divergences.

[7] Englesson E, Azizpour H. Generalized jensen-shannon divergence loss for learning with noisy labels.

**Questions:**

Why would one use your rewriting of the loss function to study the training dynamics of robust loss functions over say the gradient perspective in Wang et al. or that in the GCE paper? What novel findings do you, and can you, make only because of this framework?

Most experiments, and the only proposed fix based on the analysis, are related to MAE. As Wang et al. has studied and proposed a fix for MAE, could you clarify what novelty you bring to the understanding of MAE, and why it is significant?

Why would one use your fix for MAE over the fix proposed by Wang et al.?

---

> ### Author Response · Authors · 2023-11-20
> **Response to the weaknesses**
>
> Thanks for your incisive comments! Your questions really help make our contributions clearer in our revision:
>
> - Introduction now clearly states the core contribution
> - The derivation in section 3 is intuitively much clearer
> - The empirical results in section 4 are clearly compared to related findings.
>
> ### Regarding the weaknesses
>
> >  Q1: What is the motivation for rewriting the losses in this particular way? What novel perspective does this form give over something more intuitive like the $p_y$ and the gradient magnitude of the loss wrt to the logits? For example, in Figure 1, couldn’t one see the same thing with an x-axis with $p_y$ instead of $\Delta$, and the y-axis being the gradient magnitude wrt the logits for the MAE green curve instead of $w$?
>
> The key motivation is to unify loss functions **with different theoretical motivations** into the **same** dimensions of design choices: the sample-weighting strategy and the output regularizer, where the shared implicit loss function $\Delta(\boldsymbol{s}, y)$ provides a common ground to analyze their properties
>
> Regarding the mentioned metrics for analysis,
>
> - Although $p_y$ and $\Delta$ can both track the learning of samples, $p_y$ applies a sigmoid transform on top of $\Delta$ that ignores the change of large $\|\Delta\|$, making the change around initialization and convergence too subtle to be noticed. For example, with CE + NLS (Figure 2 f), one cannot notice the improper scale and risk of numerical overflow when viewing the distributions in $p_y$.  We discuss the benefits of $\Delta$ in the revised paragraph after Eq. (3).
>
> - Gradient magnitude can compare the loss functions **only when the shared gradient direction are extracted**, i.e., $\nabla_\boldsymbol{s} \Delta$. When regularizers are involved, e.g., MSE, our standard form helps separate the effects of regularizers, which is hard to achieve when only considering the gradient magnitude.
>
> > Q2: How isn’t it trivial that different loss functions have different gradient weights, and therefore different sample weights? If all loss functions penalized the same, there would be no reason to have different ones?
>
> Different loss functions can have different gradient **directions** and **sample weights** simultaneously. The fact that loss functions with distinct theoretical motivations share the same $\nabla_\boldsymbol{s} \Delta$ gradient component is **nontrivial**.
>
> > Q3: A single sentence motivating the inequality in Equation 5 would make it clearer.
> >
>
> In our revision, we remove the inequality in Equation 5 and add "where $\mathbb{E}_k[\Delta(\boldsymbol{s}, y)]< 0$ with standard initialization $\sigma \approx 1$ and a large $k$." right after it.
>
> > Q4: The clarity of several equations could be improved:
> >
>
> We have thoroughly revised Section 3 for clearer intuition of our derivations.
>
> > Q4.1: The equation between Equations 2 and 3 is crucial and could be made much clearer: i) a single sentence discussion or motivation for using stop gradient, ii) what $\Delta_y$ is, iii) why two additional minus signs are added in the middle equation, iv) and as $w$ is a key component of the paper, I believe it deserves a proper definition.
> >
>
> i): what we really mean is to detach $w(\boldsymbol{s}, y)$ from any computation of derivative & integral, thus preserving the same gradient in the standard form. We have rephrased "stop gradient operator" to "detach operator".
>
> ii): $\Delta_y$ is simply a typo for $\Delta(\boldsymbol{s}, y)$
>
> iii): it aims to make $w(\boldsymbol{s}, y)$ a positive function, and facilitates a direct substitution given that we do not explicitly define $w(\boldsymbol{s}, y)$
>
> iv): we dub $w(\boldsymbol{s}, y)$ the **sample-weighting function** and include its definition in our revision right after Eq. (3)
>
> > Q4.2: The equation at the end of page 3: i) why are stop gradients used in that derivation? ii) why is a factor of k and 1/k introduced?
> >
>
> i): the stop gradient issue is similar to Q4.1, i).
>
> ii): it makes the later extraction of $R_{\mathrm{NCE}}(\boldsymbol{s}) = \sum_{i=1}^{k} \frac{1}{k}\log p_i$ more straightforward. It shares similar form as confidence regularizer $R_{\mathrm{CR}}(\boldsymbol{s}) =\sum_{i=1}^{k} P(y = i) \log p_{i}$, helping the readers to observe their connection -- we can obtain $R_{\mathrm{NCE}}(\boldsymbol{s})$ by setting $P(y = i) = \frac{1}{k}$.
>
> > Q5: It was unclear to me, why the more general form that accounts for regularization was introduced, when its properties like $R(\boldsymbol{s})$ never were studied.
> >
>
> We devote the entire section 4.2 to study the effect of different $R(\boldsymbol{s})$. At the beginning of section 4, we mention that $R(\boldsymbol{s})$ constrains the distributions of $\Delta(\boldsymbol{s}, y)$ towards a predefined optimum $\Delta(\boldsymbol{s}^*, y)$ where $\boldsymbol{s}^* = \arg\min_{\boldsymbol{s}} R(\boldsymbol{s})$. In our revision, we make it more pronounced right after the introduction of the standard form in Eq. (4).

---

> ### Author Response · Authors · 2023-11-20
> **Response to Q1 of Novelty and Significance**
>
> ### Regarding Novelty and Significance
>
> > Q1: What novelty and significance does this work add over that of Wang et al. [1]?
>
> > Q1.1: What are the benefits of viewing the losses in terms of the gradient of $\Delta$ rather than the more natural gradient magnitude wrt to the logits?
>
> This question is similar to Weakness Q1 and Q2. Gradient magnitude can compare the loss functions **only when a shared gradient direction are extracted**, i.e., $\nabla_\boldsymbol{s} \Delta(\boldsymbol{s},y)$. We explicitly factorize the scale $w(\boldsymbol{s}, y)$ and direction $\nabla_\boldsymbol{s} \Delta(\boldsymbol{s}, y)$ of the gradient wrt logits $\boldsymbol{s}$ for a wide range of loss functions.
>
> When no output regularizer $R(\boldsymbol{s})$ is involved, $w(\boldsymbol{s}, y)$ and gradient magnitude are equivalent. When $R(\boldsymbol{s})$ is implicitly involved, e.g., MSE and NCE, loss functions can have different gradient directions, making comparisons of gradient magnitudes improper. In contrast, $w(\boldsymbol{s}, y)$ and $R(\boldsymbol{s})$ leads to a proper comparison among loss functions.
>
> > Q1.2: The following finding is not novel: “Zhang & Sabuncu (2018) attribute underfitting of MAE to the lack of the $1/p_y$ term in sample gradients, which “treats every sample equally” and thus hampers learning. In contrast, we show that MAE emphasizes samples with moderate $\Delta(s, y)$.” That MAE does not treat samples equally and instead focus on examples with moderate loss/$\Delta$, is not novel in this work. This is one of the main findings of Wang et al., which is clearly shown in their Figure 1 (or 2 depending on version of the paper), where examples with low and high $p_y$ have small gradients.
>
> We modify the statement to "In contrast, we show that MAE emphasizes samples with moderate $\Delta(\boldsymbol{s}, y)$ which is also observed by Wang et al. (2019a)."
>
> > Q1.3: I believe the following quote is misrepresenting the related work: “.. attribute underfitting to their low variance, making clean and noise samples less distinguishable. But as shown in Table 4 MAE can underfit data with clean labels.”. Wang et al. did not mention that the variance was low between clean and noisy samples, but rather the more general “informative” and “uninformative” examples. Furthermore, the finding that MAE underfits clean examples is not novel, and even Wang et al. clearly shows this in their Table 1.
>
> We do misinterpret their statement.  In our revision in the first paragraph of section 4.3, we rephrased our statement to "Wang et al. (2019a) view $\|\nabla_{\boldsymbol{s}} L(\boldsymbol{s}, y)\|_1$ as weights for sample gradients and argue that their low variance makes informative and uninformative samples less distinguishable. However, it is unclear how low variance of gradient magnitudes leads to underfitting." Indeed, they do not explicitly explain why low variance of gradients can lead to underfitting.
>
> We do not claim that MAE underfits clean data is novel -- it is well known in previous research.
>
> > Q1.4: Finally, Wang et al. already proposed a fix for MAE. Therefore, the novelty and significance of the proposed fix in this paper is unclear. A proper comparison (theoretically and experimentally) with other fixes for MAE is required.
>
> We should emphasize that we focus more on how our unified framework helps understanding the underfitting issue. Previous explanations have been thoroughly reviewed at the beginning of section 4.3, which are all flawed in various aspects. The proposed fixes merely aim to support the validity of our explanation:
>
> - that minimal initial sample weight leads to underfitting as shown in our analysis
> - that increasing initial sample weight indeed addresses underfitting
>
> We stress this point in the revised abstract and introduction. A comparable performance of our fixes to selected state-of-the-art results already serves our purpose. Comparisons to other fixes would be nice to have but unnecessary.
>
> We include discussions and empirical comparisons of IMAE to Wang et al. in the revised section 4.3 using our experimental settings.  See the updated Figure 4(c) for a visualization of $w(\boldsymbol{s},y)$ and Table 5,8,9 for results on CIFAR100. We find that IMAE is very sensitive to hyperparameters. Results on WebVision requires a thorough tuning of hyperparameters for IMAE, which requires more time than the rebuttal period. We will include them in later revisions.

---

> ### Author Response · Authors · 2023-11-20
> **Response to Q2 of Novelty and Significance**
>
> > Q2: What’s the significance of the proposed way of rewriting the loss functions? If one instead does similar studies in terms of loss values (e.g., Figure 3 in [1], and Figure 2 in [2]) or gradients (e.g., Theorem 1 in [3]), it seems many of the findings in the paper are already well-known. For example, that the gradients of clean examples dominates the early learning phase and then the gradients for noisy labelled examples take over, resulting in overfitting [3]. Another example, that regularization methods If there are any novel and significant findings, I believe the authors should much more clearly state, relate, and discuss the significance of them compared to related work. That MAE focuses on moderate loss examples, as shown by Wang et al. (2019a).
>
> We should again stress that:
>
> - Unifying loss functions **with different theoretical motivations** into the **same** dimensions of design choices
>
> - Connecting the design of loss functions to the design of learning curricula, which are commonly considered as distinct approaches.
>
> are theoretically **nontrivial**. These approaches are previously understood and analyzed with **distinct** approaches and motivations. Our theoretical derivations provide a **common** ground to understand and analyze them.
>
> We have summarized our key empirical findings in the last paragraph of the revised introduction:
>
> - Robust sample-weighting functions act as a sift to neglect corrupted samples
> - Regularizers can help the learning of robust loss functions
> - Underfitting is due to minimal sample weights at initializations
>
> Regarding the listed minor empirical findings:
>
> > "that the gradients of clean examples dominates the early learning phase and then the gradients for noisy labelled examples take over, resulting in overfitting" in [4]
>
> Yes, our observation is similar to [4]. They conduct theoretical analysis on linear binary classification and intuitively extend it to the multiclass setting. We remove the emphasis of this point in our revised introduction.
>
> The discussion of the relation between $\lim_{\Delta(\boldsymbol{s}, y) \to \infty} w(\boldsymbol{s}, y) = 0$ and the memoization effect is necessary for a comprehensive analysis of sample weighting functions $w(\boldsymbol{s}, y)$. We relate our results to [4] by commenting "Such explanation complements the theoretical analysis for binary linear classification in [4], which reach similar conclusion that corrupted samples dominate the expected gradients in the late training stage." in the revised last paragraph of section 4.1.
>
> > "Another example, that regularization methods If there are any novel and significant findings, "
>
> We don't understand the meaning of this question. As a speculative response, we only briefly mention that regularization preventing the memorization stage helps noise-robustness of CE in the last paragraph of section 4.1. It is included only for a more thorough discussions on the properties of $w(\boldsymbol{s}, y)$. We never claim that this observation is novel.
>
> > "That MAE focuses on moderate loss examples, as shown by Wang et al. (2019a)."
>
> In our revision, this point is clearly stated in the first paragraph of section 4.3, "In contrast, we show that MAE emphasizes samples with moderate $\Delta(\boldsymbol{s}, y)$ which is also observed by Wang et al. (2019a)."
>
> We should stress that our analysis shows that **ALL** robust loss functions has sample-weighting function $w(\boldsymbol{s}, y)$ focusing on samples with moderate $\Delta(\boldsymbol{s}, y)$ while Wang et al. (2019a) made similar observation **ONLY** for MAE. See the visualization of Figure 5 in appendix A. We mention this point in the revised caption of Table 1.

---

> ### Author Response · Authors · 2023-11-20
> **Response to experimental rigor and missing related work**
>
> ### Regarding experimental rigor
>
> > Reporting mean and standard deviation would improve the conclusiveness of the observations in the tables. Furthermore, the network predictions from several runs could be used to have more reliable histogram estimates (more data).
>
> Results in table 3-4 are reported with 3 different runs but there is no enough space to include the std. Results of WebVIsion in Table 6 are too expensive to conduct multiple random runs as stated in the caption.
>
> We would update the histogram estimates in later revisions. However, the size of training set already bears a statistically reliable estimation of the histograms.
>
> ### Regarding the missing related work
>
> > The list of robust loss functions is comprehensive, but missing some relevant ones based on information theory: i) f-divergences [5], ii) Bregman divergences [6], and iii) Jensen-Shannon divergences [7].
>
> Thanks for providing the list of related work! We include Jensen-Shannon divergences in the revised Table 2, which involves a new implicit output regularizer $R(\boldsymbol{s}) = \sum_i p_i$.
>
> Our analysis does not apply to f-divergences [5] and Bregman divergences [6] without closed-form expressions. We clearly state this in the third paragraph of the revised section 3, "We examine a broad array of loss functions with closed-form expressions in this work", and the first paragraph of the related work, "We only consider loss functions with closed-form expressions and leave others (Amid et al., 2019; Wei & Liu, 2021) to future work."

---

> ### Author Response · Authors · 2023-11-20
> **Response to Questions**
>
> ### Regarding the questions
>
> > Q1: Why would one use your rewriting of the loss function to study the training dynamics of robust loss functions over say the gradient perspective in Wang et al. or that in the GCE paper? What novel findings do you, and can you, make only because of this framework?
>
> By only considering the gradient magnitude, both Wang et al. and the GCE paper
>
> - cannot identify the effects of the implicit regularizers of loss functions, e.g., MSE, NCE
> - cannot correctly compare different loss functions. Gradient directions of the implicit regularizers makes $\|\nabla_{\boldsymbol{s}} L({\boldsymbol{s}}, y)\|_1$ of Wang et al. not comparable. Even without implicit regularizers, $\mathrm{d}L({\boldsymbol{s}}, y)/\mathrm{d} p_y$ of the GCE paper has an additional $p_y(1-p_y)$ factor in the sample gradients.
>
> Our standard form leads to proper comparison among loss functions with $w(\boldsymbol{s}, y)$ and $R(\boldsymbol{s})$.
>
>
> > Q2: Most experiments, and the only proposed fix based on the analysis, are related to MAE. As Wang et al. has studied and proposed a fix for MAE, could you clarify what novelty you bring to the understanding of MAE, and why it is significant?
>
> > Most experiments, and the only proposed fix based on the analysis, are related to MAE.
>
> In the first paragraph of the revised section 4, we explicitly state that "We mainly use MAE and CE for illustration as they exhibit typical empirical observations. Similar analysis of other loss functions are left to Appendix B."
>
> > Could you clarify what novelty you bring to the understanding of MAE, and why it is significant?
>
> Wang et al. does not provide an explicit explanation for the underfitting of MAE. Their empirical results only support the fact that their fix helps mitigate underfitting. However, their claim that "gradient magnitude’s variance matters" is not empirically supported. Based on their empirical results, one can also attribute other metrics varied due to their fix, e.g., "the expected gradient magnitude matters". Thus in the first paragraph of the revised section 4.3, we comment that "Wang et al. (2019a) view $\|\nabla_{\boldsymbol{s}} L({\boldsymbol{s}}, y)\|_1$ as weights for sample gradients and argue that their low variance makes informative and uninformative samples less distinguishable. However, it is unclear how low variance of gradient magnitudes leads to underfitting."
>
> In contrast, we support our explanation explicitly with empirical and theoretical results.
>
> - We first establish the connection between underfitting (measured by train/test accuracy) to small sample weights (measured by effective learning rate $\alpha^*$ ).
> - We then theoretically connects the increased number of classes to the decreased $\Delta(\boldsymbol{s},y)$ and sample weights at initialization, and empirically support it with results in Figure 3.
> - We finally establish the causal relation between minimal initial sample weights and underfitting. The fact that increased initial sample weights with our fixes eliminates underfitting support this claim well.
>
> > Q3: Why would one use your fix for MAE over the fix proposed by Wang et al.?
>
> Our fix are less sensitive to the hyperparameter $\tau$: the same $\tau = 3.4$ works well for a wide range of noise rates with MAE, while IMAE Wang et al. requires careful hyperparameter tuning for each noise setting (stated in the appendix D of their paper). In addition, under the same settings, our fix achieves better results than IMAE.
>
> Nonetheless, we recommend designing new learning curricula based on our framework and empirical findings instead of sticking with the classic but flawed MAE. As loss functions can be viewed as sample-weighting curricula, our findings suggest plenty of approaches to design good curricula. As a quick list:
>
> 1. Blending the implicit curricula of robust loss functions to those explicitly designed, e.g., approach proposed in "DivideMix: Learning with Noisy Labels as Semi-supervised Learning"
> 2. We can use other metrics than $\Delta(\boldsymbol{s}, y)$ to identify the constantly unlearned samples, e.g., $\Delta(\boldsymbol{s}, y) - m$  where $m$ is a running average of $\Delta(\boldsymbol{s}, y)$, and assign small weights to small $\Delta(\boldsymbol{s}, y) - m$, which can avoid the underfitting issue due to minimal initial sample weights.
> 3. Use a proper regularizer for a good $\Delta(\boldsymbol{s}^*, y)$ with $\boldsymbol{s}^* = \arg\min_{\boldsymbol{s}} R(\boldsymbol{s})$ to facilitate learning with a large number of classes and avoid overfitting noisy samples. Existing confidence reducing regularizers all share the same minimum with $\Delta(\boldsymbol{s}^*, y) = - \log k$ where $k$ is the number of classes, which may risk underfitting with robust loss functions under large $k$.
> 4. Utilize the sample sifting effect of robust loss functions to **select** corrupted samples and either drop them or utilize semi-supervised learning to obtain surrogate labels.

---

> ### Comment · Reviewer_L9A1 · 2023-12-03
> **Thank you for the detailed rebuttal!**
>
> I would like to thank the authors for their detailed rebuttal, which addressed some, but unfortunately not, my main concern: What advantages does the split in terms of $\Delta$ and $R(\boldsymbol{s})$ (Equations 3 and 4) have over the full ($\frac{\partial L}{\partial \boldsymbol{s}}$) gradient perspective?
>
> The logic of the argument the authors raise is as follows
>
> 1. Without implicit regularisation, then the perspectives are equivalent.
> 2. With implicit regularisation, the gradient directions can differ.
> 3. If gradient directions differ, then it is improper to compare gradient magnitudes, but comparing $w(\boldsymbol{s},y)$ and $R(\boldsymbol{s})$ is proper.
>
> As there is no difference without implicit regularisation, we focus on the regularisation case. Unfortunately, the authors neither give a reason for why comparing gradient magnitudes is improper in this case, nor why comparing $w(\boldsymbol{s},y)$ and $R(\boldsymbol{s})$ is proper. Therefore, I look into it myself.
>
> Without regularisation, the loss functions can be written as
>
> $L’(\boldsymbol{s},y) = -w(\boldsymbol{s},y) \cdot \Delta(\boldsymbol{s},y)$
>
> where all loss functions share the same $\Delta(\boldsymbol{s},y)$ and thus $\nabla_{\boldsymbol{s}}\Delta(\boldsymbol{s},y)$ as well. Thus, it makes sense to compare gradient magnitudes between different losses in terms of $w(\boldsymbol{s},y)$, as it is the only thing that differs. However, in the case of regularisation the “standard form” is
>
> $L’’(\boldsymbol{s},y) = -w(\boldsymbol{s},y) \cdot \Delta(\boldsymbol{s},y) + \lambda(\boldsymbol{s}, y) R(\boldsymbol{s})$
>
> Here things get complicated. Once again, $\Delta(\boldsymbol{s},y)$ is shared between different losses but now there are three parts that differ: $w(\boldsymbol{s},y)$, $\lambda(\boldsymbol{s}, y)$ and $R(\boldsymbol{s})$.
>
> It is unclear to me how this form makes it possible to compare losses in a proper way in terms of $w(\boldsymbol{s},y)$ and $R(\boldsymbol{s})$ as claimed (although not explained) by the authors. For example, as $R(\boldsymbol{s})$ differs between losses, so do the gradient directions, and therefore any comparisons in terms of $w(\boldsymbol{s},y)$ should have the same problems as full gradient perspectives. The only case I can think of where comparisons make sense, is when $R(\boldsymbol{s})$ is the same for the losses under study, then what differs are $w(\boldsymbol{s},y)$ and $\lambda(\boldsymbol{s}, y)$ and we can study these. Note that, in this case, the gradient directions are the same, so we could also perform gradient magnitude comparisons with the full gradient approach.
>
> To me, to have a general comparison of gradients, one has to take both magnitude and directions into account (can be done in full gradient approach), e.g., in terms of cosine similarity. Splitting the gradient into different factors and not taking all parts into consideration, currently, only seem to either lead to improper comparisons or only being able to study special cases. I hope the authors are able to extract useful information from their proposed perspective, but as of now, it is unclear how the perspective improves our understanding over the full gradient approach. Furthermore, I see no reason why the main observations: "Robust sample-weighting functions act as a sift to neglect corrupted samples" and "Underfitting is due to minimal sample weights at initializations", could not be observed with the full gradient magnitude approach.
>
> For these reasons, I still believe the work is marginally below the acceptance threshold.

---

### Official Review · Reviewer_CdpW · 2023-11-06

**Soundness:** 3 good
**Presentation:** 3 good
**Contribution:** 3 good
**Rating:** 6
**Confidence:** 3

**Summary:**

The authors examine the challenges of underfitting and the factors influencing the robustness of loss functions in the context of training with noisy labels. They approach these questions by analyzing robust loss functions through a lens that emphasizes the importance of sample-weighting strategies and an optional implicit regularizer. To address underfitting, they suggest modifying these sample-weighting approaches. Additionally, they present evidence that refining the schedule of learning rate adjustments can enhance the robustness of the loss functions.

**Strengths:**

* This work connects several popular robust loss designs to a sample-weighting curriculum.

* Empirically, the authors explain the two open questions in the literature of learning with noisy labels. The introduce of a marginal effective learning rate looks interesting and helps with explaining the underfitting issue. And the shifting of soft-margin mitigates the underfitting, especially when the number of classes is large.

**Weaknesses:**

* The presentation of experiments could be further improved, i.e., what is $\tau$ in Figure 4.

* The proposed strategy for shifting and rescaling appears to hold potential; however, its design is somewhat heuristic and depends heavily on a crucial hyper-parameter. This may hinder the efficient usage of the proposed method in practice.

**Questions:**

My main concerns are from the empirical sections:

* what is $\tau$ in Figure 4?

* In Table 3, why MAE has such pretty bad performances under CIFAR-100?

* Maybe I missed some important details, I was wondering how authors pick  $\tau$ for reporting experiment results in Table 5, 6.

---

> ### Author Response · Authors · 2023-11-20
>
> Thanks for your comments! We have carefully revised our submission for better clarity.
>
> ### Regarding the weakness
>
> > Q1: The presentation of experiments could be further improved, i.e., what is $\tau$ in Figure 4.
>
> $\tau$ is the hyperparameter of the shifted/scaled weighting function $w^{+}(\boldsymbol{s}, y)$ and  $w^{*}(\boldsymbol{s}, y)$. It is denoted in the caption of Figure 4 in our revision
>
> > Q2: The proposed strategy for shifting and rescaling appears to hold potential; however, its design is somewhat heuristic and depends heavily on a crucial hyper-parameter. This may hinder the efficient usage of the proposed method in practice.
>
> In fact, we deliberately keep our fixes simple instead of more intricate designs. The scaling&shifting fixes aim to show that by avoiding minimal initial samples weights, underfitting loss functions can become as performant as previous state-of-the-art results, thus supporting our explanation of the underfitting issue.
>
> We do not claim for a general well-performing approach in the present work that focus on comprehensively understanding existing loss functions, which can facilitate the design of better loss functions and learning curricula in future work. We have stressed these points in our revised Introduction.
>
> A good hyperparameter $\tau$ can be generally applicable to most noise settings as shown by our results. It only requires a mild adjustment when changing tasks with different difficulties.
>
> ### Regarding the questions
> > Q1: what is $\tau$ in Figure 4?
>
> See response for weakness Q1
>
> > Q2: In Table 3, why MAE has such pretty bad performances under CIFAR-100?
>
> It is due to the minimal sample weights that $w_{\mathrm{MAE}}$ assign to samples at initialization, which has been thoroughly discussed in section 4.3:
> - the first paragraph provide a quick review of existing observations and explanations
> - the second confirms and intuitively explains it
> - the third reveals the cause with increased number of classes $k$ and how it leads to minimal initial sample weights
> - the fourth provide fixes to address it, further confirming that underfitting is indeed caused by minimal initial sample weights
>
> > Q3: Maybe I missed some important details, I was wondering how authors pick $\tau$ for reporting experiment results in Table 5, 6.
>
> $\tau$ in Tables 5,8,9 is tuned on CIFAR100 with symmetric noise rate $\eta = 0.4$. Due to the excessive training time required to train on WebVision, $\tau$ in Table 6 is roughly tuned for each setting based on the observation that smaller $\tau$ helps avoid underfitting but undermines noise robustness.

---

### Meta-Review · Area_Chair_GUqQ · 2023-12-18

**Metareview:**

An interesting take on loss function, but which falls short of achieving its goal.

Any such take on a key parameter of learning algorithms like the loss function has to generally be based on first principles (explanation: https://www.di.ens.fr/~fbach/ltfp_book.pdf). Otherwise, the approach takes the risk of being dependent on a bias that holds only for the subset selected and for reasons specific to this subset, which can even be detrimental to the analysis and the broader understanding.

The paper investigates losses for class probability estimation. The basic first principle approach to these losses can be found here: https://dl.acm.org/doi/10.5555/1756006.1953012, which can be extended to multiclass (e.g. here: https://proceedings.neurips.cc/paper/2011/hash/2afe4567e1bf64d32a5527244d104cea-Abstract.html). When data is subject to general corruption (not just the one mentioned in the paper), the analysis can be tackled this way: https://proceedings.mlr.press/v162/sypherd22a.html.

What reviewer L9A1 is pushing for is a broadening of the analysis, which is a very fair point given above. The versatility adovcaed for by reviewer pjXu takes on the experimental standpoint but is very much aligned.

I am curious to see as to whether the approach claimed by the authors can be derived from first principles. I believe it is worth trying *and* it is mandatory for the paper to pass the acceptance bar for a venue like ICLR.

**Justification For Why Not Higher Score:**

As it stands, very marginal take on loss functions.

**Justification For Why Not Lower Score:**

N/A

---

### Decision · Program_Chairs · 2024-01-16

Reject